# BACK PROPAGATION THROUGH AUCTIONS: FIRST-ORDER POLICY GRADIENT FOR AUTO-BIDDING

## ABSTRACT

In online advertising, auto-bidding agents compete in high-frequency auctions by setting a bidding parameter for each time interval, that scales estimated impression values into actual bids. While prior work has framed this sequential decision problem as a reinforcement learning (RL) task, we identify that standard RL methods overlook key structural properties of the auto-bidding environment: agents receive fine-grained, impression-level feedback, and the objective is nearly differentiable due to the high density of impressions within each interval. We leverage this structure to propose First-Order policy gradient for auto-Bidding (FOB), a method that directly computes policy gradients by smoothing historical auction data and backpropagating through the sequential auctions. FOB leverages Myerson's lemma, a cornerstone of auction theory, to explicitly derive gradients. We validate FOB on AuctionNet, a public auto-bidding environment, where it consistently outperforms standard RL baselines and domain-specific auto-bidding methods, achieving superior performance with greater stability and faster convergence.

## 1 INTRODUCTION

In modern online advertising, ad impressions are allocated through auctions—the core economic engine of e-commerce, social media, and search platforms (Aggarwal et al., 2024). Impressions arrive sequentially and unpredictably, varying in timing, volume, and user context, often at rates of millions per minute. To compete effectively in such high-frequency auctions, advertisers rely on auto-bidding agents provided by the platforms (Google; Meta). The goal of auto-bidding is to maximize the total value of impressions won over a fixed campaign period (e.g., one day), while respecting the advertiser's budget constraint.

A theoretically grounded and widely adopted tool for auto-bidding is *the optimal bidding formula* (Balseiro et al., 2015; Aggarwal et al., 2019; 2024): Bid proportionally to each impression's estimated value, scaled by a single parameter. In practice, considering the dynamic auction environment, platforms implement this formula by dividing the campaign period into discrete time steps (e.g. 48 half-hour segments per day), and assigning one bidding parameter per step. All impressions arriving within that step are then bid on using the same parameter. This step-level framework elegantly balances performance and practicality: it is simple to implement, computationally efficient at scale, and responsive enough to adapt to real-time environment changes. As a result, it has become the de facto framework in industry (Wu et al., 2018; Gao et al., 2022; Ou et al., 2023; Chen, 2025).

Consequently, the auto-bidding problem reduces to a *step-level* decision-making task: Dynamically selecting the optimal bidding parameter for each step based on the observed information. A rich body of algorithmic work has emerged to tackle it. Early approaches used heuristic rules (Lee et al., 2013; Geyik et al., 2016) or classical control methods like PID controllers (Yang et al., 2019; Karlsson, 2020; Zhang et al., 2022). More recently, reinforcement learning (RL) has emerged as a promising paradigm for auto-bidding (Wu et al., 2018; He et al., 2021; Mou et al., 2022; Li et al., 2024; Korenkevych et al., 2024), modeling the problem as a Markov Decision Process (MDP) to directly optimize long-term value. However, the auto-bidding environment is inherently highly stochastic (Lu et al., 2019): impression volumes, values, and competition landscapes fluctuate across time steps. This poses significant challenges for standard RL algorithms, which often suffer from poor sample efficiency and unstable convergence.

In this work, we identify a fundamental insight: *Auto-bidding possesses structural properties that are fundamentally richer than standard MDPs.* Unlike typical RL environments, where agents observe only scalar rewards and state transitions, the auto-bidding environment provides complete impression-level feedback: after every auction, we observe both the impression's value and the winning price (i.e., the minimum bid needed to win). This enables counterfactual policy evaluation, i.e., the ability to simulate, on historical logged instances, the exact outcome of any bidding policy. Moreover, due to the high volume of impression arrivals in each step, the mapping from bidding parameters to step-level outcomes is highly smooth: Small increases or decreases in the bidding parameter result in incremental wins or losses over a subset of impressions, leading to near-continuous changes in both reward and budget consumption. Crucially, this allows us to estimate how an infinitesimal adjustment to the policy parameters affects the objective function. This yields a *first-order policy gradient* (Heess et al., 2015; Suh et al., 2022), a direct, low-variance signal for policy optimization.

Building on these structural insights, we propose First-Order policy gradient for auto-Bidding (FOB). For each time step, we construct smooth, differentiable approximations of the step-level reward and cost functions, which enables direct gradient computation with respect to the action. We then backpropagate from the last step to the first, to obtain the gradient of the objective on policy parameters. Crucially, FOB leverages Myerson's Lemma (Myerson, 1981), a foundational result in auction theory, to analytically derive the gradient of cost from the gradient of reward. FOB also includes a specialized approximation scheme for handling early budget depletion. The result is a simple and stable algorithm that requires no value networks, no temporal difference learning, and minimal hyperparameter tuning, yet achieves faster convergence and higher performance than standard zeroth-order RL methods in highly dynamic auto-bidding environments. Conceptually, FOB separates the challenge of learning the highly stochastic and complicated impression arrival distribution from the task of optimal planning within a single, deterministic instance.

Our key contributions in this work are:

- We study the auto-bidding problem under an MDP formulation, and reveal fundamental structural properties: the availability of rich, impression-level feedback, and the smoothness of action-to-reward and action-to-cost mappings under high impression density.

- We exploit this structure and propose FOB, a stable and efficient policy gradient estimator for auto-bidding. FOB finds the direction for policy improvement by smoothing historical instances and backpropagating through auctions. The resulting algorithm is remarkably simple, requiring no value networks, no TD learning, and minimal hyperparameter tuning.

- We evaluate FOB on AuctionNet (Su et al., 2024), a public auction environment derived from real-world ad platforms. FOB consistently outperforms standard RL baselines and domain-specific auto-bidding methods across different budget levels.

## 2 PRELIMINARIES

**Notations.** We use $\mathbb{I}\{\cdot\}$ to denote the indicator function. We use $\mathbb{R}_+$ to denote the set of non-negative real numbers. We use $\mathbb{N}_+$ to denote the set of non-negative integers. For $N \in \mathbb{N}$, we use $[N]$ to denote $\{1, 2, \cdots, N\}$. For any set $\mathcal{S}$, we use $\Delta(\mathcal{S})$ to denote the set of probability distributions over $\mathcal{S}$.

### 2.1 ONLINE ADVERTISING AUCTIONS

In online display advertising, advertisers participate in auctions to acquire ad impressions. During a time period (e.g., one day), suppose there are $n \in \mathbb{N}_+$ impressions arriving sequentially and indexed by $i \in [n]$, each of which triggers an ad auction among advertisers. We focus on a prevalent auction mechanism, i.e., *second-price auction* (Aggarwal et al., 2024), where all advertisers submit their bids simultaneously, the one with the highest bid wins the impression, and pays the second-highest bid. From an advertiser's perspective, each second-price auction is characterized by two parameters: the predicted value $v_i \in \mathbb{R}_+$ of the impression (e.g. the click-through rate, the conversion rate, etc.), estimated in real-time via machine learning models (Zhou et al., 2018) based on user features, item features, and historical interaction data; and the winning price $p_i \in \mathbb{R}_+$, i.e., highest bid among other bidders. Before bidding, the advertiser knows $v_i$ but not $p_i$. If her bid $b_i > p_i$, the advertiser wins, obtains the impression with value $v_i$, and pays $p_i$; otherwise, they lose and pay nothing.

## 2.2 OPTIMAL BIDDING FORMULA

A common objective in auto-bidding is to maximize the total value of impressions won, subject to a budget constraint $B \in \mathbb{R}_+$. For $n$ impression opportunities, each with an estimated value $v_i$ and winning price $p_i$, the goal is formulated as

$$\max_{b_1, \cdots, b_n} \sum_{i=1}^n v_i \mathbb{I}\{b_i > p_i\}, \text{ subject to } \sum_{i=1}^n p_i \mathbb{I}\{b_i > p_i\} \leq B. \tag{1}$$

It is well-established in the literature (Balseiro et al., 2015; Aggarwal et al., 2019; 2024) that, for this auto-bidding problem, there exists an optimal bidding formula $b_i = a \cdot v_i$. That is, an agent can achieve optimal performance by bidding in proportion to the impression values. The optimal bidding parameter $a$ depends on all values $\{v_i\}_{i=1}^n$ and winning prices $\{p_i\}_{i=1}^n$ throughout the time period, and can be solved by linear program solvers. In Appendix A.1, we provide a self-contained proof of this result, with the explicit form of parameter $a$.

**Step-level auto-bidding.** In real-world advertising systems, however, the auction environment is dynamic and non-stationary (Liang et al., 2023). The impression traffic, values, and competitors' bids may vary significantly over time, making it impractical to predict all $\{v_i\}_{i=1}^n$ and $\{p_i\}_{i=1}^n$, and compute a fixed optimal $a$. To address this, practitioners adopt a step-level decision framework: The time horizon (e.g., one day) is partitioned into $T$ discrete steps (e.g., 48 half-hour intervals). At the start of each step $t$, the agent selects a step-specific bidding parameter $a_t$, which is then applied to all impressions arriving within that step. This framework has become the de facto standard in many industrial auto-bidding systems (Gao et al., 2022; Ou et al., 2023; Chen, 2025). The resulting *step-level auto-bidding problem* is then finding the optimal online policy to decide the step-level bidding parameters. See Appendix B for a review on previous works under this framework.

## 2.3 REINFORCEMENT LEARNING

An MDP (Bellman, 1957; Puterman, 2014) is characterized by four components: a state space $\mathcal{S}$, an action space $\mathcal{A}$, transition dynamics $\mathcal{P} : \mathcal{S} \times \mathcal{A} \to \Delta(\mathcal{S})$, and a reward function $r : \mathcal{S} \times \mathcal{A} \to \mathbb{R}$.[1] The objective of reinforcement learning (RL) is to identify a policy $\pi_\theta : S \to \Delta(A)$ that maximizes the expected cumulative reward $J(\pi_\theta) = \mathbb{E}_{\pi_\theta}\left[\sum_{t=1}^T r(s_t, a_t)\right]$, where the expectation depends on both the policy and the environment's transition dynamics.

## 2.4 STOCHASTIC GRADIENT ESTIMATION

The RL problem is essentially maximizing an expectation, $\mathbb{E}_{p(z;\theta)}[y(z)]$, with respect to parameters $\theta$ defining the distribution under the expectation. In RL, $z$ represents a trajectory, $y(z)$ is the return, and $p(z; \theta)$ is the trajectory distribution induced by the policy $\pi_\theta$. The key is obtaining Monte Carlo estimators (Mohamed et al., 2020) for the gradient $\nabla_\theta \mathbb{E}_{p(z;\theta)}[y(z)]$. We introduce two commonly used estimators.

**Zeroth-Order (Score Function) Gradient Estimator.** The REINFORCE estimator (Williams, 1992; Sutton et al., 1999), a cornerstone of policy gradient methods in RL, is a zeroth-order estimator which estimate gradients using only samples of function value:

$$\nabla_\theta \mathbb{E}_{p(z;\theta)}[y(z)] = \mathbb{E}_{p(z;\theta)}[y(z)\nabla_\theta \log p(z; \theta)].$$

This estimator is straightforward to compute for parameterized distributions. However, the variance is often high, and the success of zeroth-order policy gradient methods relies on various variance-reduction techniques (Schulman et al., 2015; 2017) and implementation tricks (Huang et al., 2022).

**First-Order (Pathwise) Gradient Estimator.** First-order methods exploit structural knowledge of how randomness propagates through the system. When $z$ can be expressed as a deterministic, differentiable transformation $z = g(\epsilon; \theta)$ of a parameter-independent noise source $\epsilon \sim p(\epsilon)$, the Law of the Unconscious Statistician (LOTUS) Grimmett & Stirzaker (2020) yields $\mathbb{E}_{p(z;\theta)}[y(z)] =$

---

[1]This paper focuses on episodic MDPs with a discount factor of 1.

$\mathbb{E}_{p(\epsilon)}[y(g(\epsilon;\theta))]$. This identity pushes the parameter $\theta$ to the inside objective function, making the distribution under the expectation free of the parameters. This enables the gradient estimator

$$\nabla_\theta \mathbb{E}_{p(z;\theta)}[y(z)] = \mathbb{E}_{p(\epsilon)}[\nabla_\theta y(g(\epsilon;\theta))].$$

As noted by Ghadimi & Lan (2013); Mohamed et al. (2020), the first-order estimator often results in much less variance compared to the zeroth-order one, which leads to faster and more stable convergence. However, it relies on a known differentiable function $g$ that transforms the raw random source into the objective. In the context of RL, this requires a model of the environment transition and rewards which is differentiable with respect to actions. Previous works have developed physics-based differentiable simulators for robotics (Xu et al., 2022; Xing et al., 2025), or trained neural networks to model environments (Heess et al., 2015; Clavera et al., 2020), both of which require effort and may introduce gradient biases. Our work suggests that in auto-bidding, a reliable differentiable model is almost free to obtain: We only need a number of historical instances. See Appendix B for more discussion on related work.

## 3  MODEL

We start by formalizing the step-level auto-bidding problem. Consider an auto-bidding agent, bidding for an advertiser with budget $B \in \mathbb{R}_+$. There are $T \in \mathbb{N}_+$ time steps in an episode. Let $B_t$ denote the remaining budget at the start of step $t$, with $B_1 = B$. In step $t \in [T]$:

1. If remaining budget $B_t > 0$, the agent decides $a_t \in \mathbb{R}_+$, the bidding parameter for the current step under the linear bidding formula.

2. A sequence of $n_t$ impressions arrives, where $n_t \leq N$, and $N \in \mathbb{N}_+$ is an upper bound on the impression number in any round. Each impression $i \in [n_t]$ is characterized by a value $v_{t,i} \in \mathbb{R}_+$ and a price $p_{t,i} \in \mathbb{R}_+$. For each impression $i$:

   • The agent observes the value $v_{t,i}$, and submits a bid $b_{t,i} = a_t \cdot v_{t,i}$.
   • If $b_{t,i} > p_{t,i}$, i.e., the agent's bid is the highest among all bidders, the agent wins the auction, obtains value $v_{t,i}$, pays the price $p_{t,i}$ and updates the remaining budget.

3. The bidding process stops immediately if the budget is depleted at any point.

**Step-level Reward and Cost.** The result of bidding in each step is obtaining value and consuming budget. When defining the step-level reward and cost, we should be careful in dealing with budget depletion at the middle of a step. Let $x_{t,i} = \mathbb{I}\{a_t \cdot v_{t,i} > p_{t,i}\} \in \{0,1\}$ indicate whether the agent's bid can win impression $i$ in step $t$. The *intended* step-level reward and cost, assuming sufficient budget, is then $r_t^{\text{int}}(a_t) = \sum_{i=1}^{n_t} v_{t,i} x_{t,i}$, and $c_t^{\text{int}}(a_t) = \sum_{i=1}^{n_t} p_{t,i} x_{t,i}$. The *actual* reward and cost depend on the relationship between the intended cost and the remaining budget $B_t$: When the budget is sufficient for the intended cost, i.e., $B_t \geq c_t^{\text{int}}(a_t)$, the agent spends that cost $c_t = c_t^{\text{int}}(a_t)$ and gets reward $r_t = r_t^{\text{int}}(a_t)$. Then the budget consumes by $B_{t+1} = B_t - c_t$. If the budget is in-

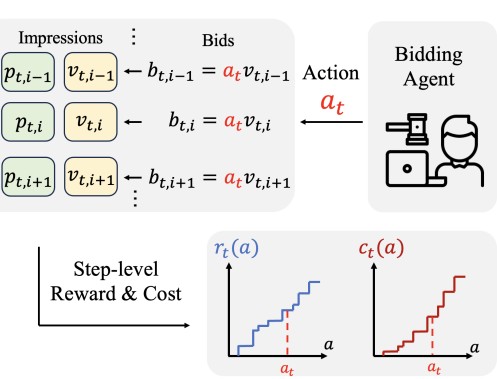

Figure 1: Illustration of the auto-bidding process within a single (non-depletion) step $t$.

sufficient for the intended cost, i.e., $B_t < c_t^{\text{int}}(a_t)$, the agent can only buy part of the intended impressions. Let $j$ be the largest integer such that $\sum_{i=1}^{j} p_{t,i} x_{t,i} \leq B_t$. Then we have $r_t = \sum_{i=1}^{j} v_{t,i} x_{t,i}$, and for simplicity, we set $c_t = B_t$ to clear the budget. We call such step the *depletion step*.

We assume *stochastic arrival* of requests. Let $\gamma_t = (n_t, \{v_{t,i}\}_{i=1}^{n_t}, \{p_{t,i}\}_{i=1}^{n_t})$ summarize the information of all impressions arriving in step $t$. We assume that $\gamma_t$ is a random variable with distribution $P_t$. Combining all steps, we define the *instance* for one episode to be $\mathcal{I} = (\gamma_1, \cdots, \gamma_T)$. Then denote the probability distribution of $\mathcal{I}$ by $P$.

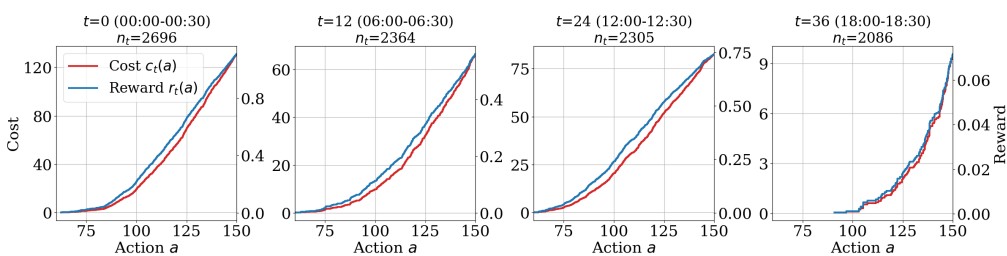

Figure 2: Reward and cost functions across time steps. Each subplot shows the reward $r_t(a)$ (blue) and cost $c_t(a)$ (red) functions at different time steps $t \in \{0, 12, 24, 36\}$. The data is collected from AuctionNet (Su et al., 2024).

The step-level auto-bidding problem can be modeled by the following Markov Decision Process (MDP). All the information observable till step $t$ is encoded in the state $S_t = (\{\{v_{\tau,i}\}_{i=1}^{n_\tau}\}_{\tau=1}^{t-1}, \{c_\tau\}_{\tau=1}^{t-1}, \{a_\tau\}_{\tau=1}^{t-1})$. The agent's *policy* is $\pi_\theta$, mapping from a state to a probability distribution over actions. Running a policy $\pi_\theta$ on an instance $\mathcal{I}$ produces return $J(\pi_\theta; \mathcal{I}) = \sum_{t=1}^{T} r_t$. We denote the expected return by $J(\pi_\theta) = \mathbb{E}_{\mathcal{I} \sim P}[J(\pi_\theta; \mathcal{I})]$. The agent's goal is to find a policy that maximizes the expected return, i.e., $\max_\theta J(\pi_\theta)$.

**State Space.** Directly using the high-dimensional state $S_t$ poses significant challenges for RL algorithms, thus state abstraction is required. Previous works (Wu et al., 2018; He et al., 2021; Mou et al., 2022) have not reached an agreement on the state space design principle for auto-bidding. In this work, we adopt a minimalist three-dimensional state: $s_t = (t, B_t, B)$, i.e., the current step, the remaining budget, and the total budget. We theoretically justify this choice:

**Theorem 3.1.** *Assume that the requests $\gamma_1, \cdots, \gamma_T$ are mutually independent. Consider a new MDP with state $s_t = (t, B_t, B)$. The optimal policy in this MDP is also optimal in the original MDP with the full history state $S_t$.*

The theorem guarantees the optimality of this state abstraction under the assumption of independent arrivals across steps. We provide a proof in Appendix A.2, by showing that the mapping $\phi(S_t) = s_t$ is a model-irrelevance abstraction (Li et al., 2006). While we present our work with this minimal representation, our proposed method naturally extends to richer state spaces.

## 4 METHOD

In this section, we introduce our approach for the step-level auto-bidding problem under the MDP formulation. Our method is natural given the following structural properties of the problem.

### 4.1 MOTIVATION

Under the MDP formulation, existing works address the auto-bidding problem directly borrowing existing deep RL algorithms (Wu et al., 2018; Zhao et al., 2018; Mou et al., 2022; Li et al., 2024). However, we find that auto-bidding has a rich feedback structure that standard RL methods fail to exploit. In standard RL frameworks, when an agent takes action $a_t$ in state $s_t$, it receives an immediate reward $r_t$ and the next state $s_{t+1}$. In contrast, auto-bidding provides fine-grained, impression-level information: for each individual impression $i$ within step $t$, the agent observes its value $v_{t,i}$ and price $p_{t,i}$ . [2] This yields a complete instance $\mathcal{I} = \{(n_t, \{v_{t,i}\}_{i=1}^{n_t}, \{p_{t,i}\}_{i=1}^{n_t})\}_{t=1}^{T}$, which provides full knowledge of one realization of the environment.

Given such an instance $\mathcal{I}$, we can simulate any policy $\pi_\theta$, compute its wins and losses across all auctions, and evaluate its return $J(\pi_\theta; \mathcal{I})$. Crucially, this enables off-policy evaluation of arbitrary policies using historical data, not only the policy that has generated the data, as in standard (offline)

---

[2]The prices of winning impressions are always observed, since the agent's budget is reduced by that amount. We assume that the prices of losing impressions are also revealed by the platform (auctioneer) after the auction. This assumption is standard in the literature (He et al., 2021; Balseiro et al., 2023) and reflects real-world practice: major platforms often train bidding agents on behalf of advertisers (Decarolis et al., 2020; Wen et al., 2022; Chen et al., 2023). We discuss extension beyond this assumption in Appendix C.

RL. Moreover, the following observation renders historical instances more useful than just policy evaluation, and allows us to move from zeroth-order to first-order.

**Observation 1.** In most auto-bidding scenarios, impression arrivals are dense, i.e., the number of impressions per time step is large. Moreover, the price of each individual impression is typically very small compared with the total budget.

For example, in AuctionNet (Su et al., 2024) which simulates Alibaba's real-world bidding environment, each episode (48 time steps) contains over $10^5$ auction requests, with more than $10^3$ requests per step. An advertiser typically wins $10^2 \sim 10^4$ impression opportunities each episode. Similar arrival density is reported in Yuan et al. (2013); Khirianova et al. (2025). The "small individual price" observation also aligns with the standard "small bid" or "large budget" assumption in online resource allocation (Mehta, 2013; Devanur et al., 2019; Roth, 2023). This high impression density induces smooth reward and cost functions, as shown in Figure 2. We are thus able to evaluate how the reward $r_t$ and cost $c_t$ will change under any infinitesimal lift or drop on the bidding parameter $a_t$. This provides *first-order* information for policy optimization.

We summarize three key distinctions between the auto-bidding problem and standard MDPs, which directly motivate our design:

1. The randomness of the environment only stems from the stochastic arrival of impressions, i.e., $\mathcal{I} \sim P$. We typically have access to a number of historical instances $\mathcal{I}$ (bidding logs), offering rich knowledge on the distribution $P$.

2. For any fixed instance $\mathcal{I}$, the system is deterministic, with *explicit formulas* for transitions and rewards (given in Section 3). This allows us to evaluate $J(\pi_\theta; \mathcal{I})$ of *any* bidding policy $\pi_\theta$.

3. Under high impression density (Observation 1), rewards and costs are smooth, making $J(\pi_\theta; \mathcal{I})$ *nearly differentiable* with respect to the policy parameters $\theta$.

On each historical instance $\mathcal{I}$, one could evaluate how the return $J(\pi_\theta; \mathcal{I})$ would change under an infinitesimal disturbance to the policy parameters $\theta$, and find the direction that leads to the fastest ascent of the return. Formally, we leverage the first-order gradient estimator: $\nabla_\theta J(\pi_\theta) = \mathbb{E}_{\mathcal{I}}[\nabla_\theta J(\pi_\theta; \mathcal{I})]$. In the following subsection, we derive the explicit form of $\nabla_\theta J(\pi_\theta; \mathcal{I})$.

### 4.2 THE FIRST-ORDER GRADIENT ESTIMATOR

For some policy $\pi_\theta$ and instance $\mathcal{I}$, our goal is to evaluate $\nabla_\theta J(\pi_\theta; \mathcal{I})$. Since the return is the sum of step-level rewards, we require $\nabla_\theta r_t$ for all $t$. We distinguish between two types of steps: In a non-depletion step $t \in [T]$, the reward $r_t(a_t) = \sum_{i=1}^{n_t} v_{t,i} \mathbb{I}\{a_t \cdot v_{t,i} > p_{t,i}\}$ only depends on $a_t$, not on the state $s_t$. On the other hand, in a depletion step $u \in [T]$, the budget is exhausted in the middle of the step, so the reward $r_u$ highly depends on the remaining budget $B_u$ at the start of the step. We first address non-depletion steps.

**Non-depletion Step: Gradients on Actions.** The reward function $r_t(a) = \sum_{i=1}^{n_t} v_{t,i} \mathbb{I}\{a \cdot v_{t,i} > p_{t,i}\}$ is a non-decreasing, piecewise-constant function in $a$. Its true gradient is zero almost everywhere. However, under Observation 1, we approximate it with a smoothed, differentiable surrogate $\tilde{r}_t(a)$, and obtain $\nabla_a \tilde{r}_t$. We present two practical smoothing strategies:

- *Piecewise-Linear Approximation:* The breakpoints of the reward function occur precisely at the values $a = p_{t,i}/v_{t,i}$ for each impression $i$. We construct $\tilde{r}_t(a)$ by linearly interpolating between consecutive points in $\{(p_{t,i}/v_{t,i}, r_t(p_{t,i}/v_{t,i}))\}_{i=1}^{n_t}$ with line segments. Let $a_{(1)}, \cdots, a_{(n_t)}$ denote the sorted breakpoints. For $a \in [a^{(k)}, a^{(k+1)})$, the gradient is then given by: $\nabla_a \tilde{r}_t(a) = (r_t(a^{(k+1)}) - r_t(a^{(k)}))/(a^{(k+1)} - a^{(k)})$.

- *Savitzky-Golay (SG) Filter:* Following Savitzky & Golay (1964), we fit a local quadratic function to the breakpoint data $\{(p_{t,i}/v_{t,i}, r_t(p_{t,i}/v_{t,i}))\}_{i=1}^{n_t}$ around $a$. The derivative of the fitted quadratic provides the desired gradient $\nabla_a \tilde{r}_t(a)$.

**Auction Theory Provides Analytical Cost Gradients.** To obtain the gradient of the cost $\nabla_a \tilde{c}_t$, the above smoothing techniques can be performed similarly on the cost function $c_t(a) = \sum_{i=1}^{n_t} p_{t,i} \mathbb{I}\{a v_{t,i} > p_{t,i}\}$. However, we derive a more efficient and theoretically consistent method by leveraging a fundamental identity from auction theory.

**Proposition 4.1.** *For the reward function $r_t(a) = \sum_{i=1}^{n_t} v_{t,i} \mathbb{I}\{a \cdot v_{t,i} > p_{t,i}\}$ and the cost function $c_t(a) = \sum_{i=1}^{n_t} p_{t,i} \mathbb{I}\{a \cdot v_{t,i} > p_{t,i}\}$. The following identity holds, $c_t(a) = ar_t(a) - \int_0^a r_t(\alpha) d\alpha$.*

This identity follows from Myerson's lemma (Myerson, 1981), a foundational result in auction mechanism design that characterizes the unique payment rule for single-parameter truthful mechanisms. The proof is deferred to Appendix A.3. [3] To ensure consistency between our smoothed functions, we enforce $\tilde{c}_t(a) = a\tilde{r}_t(a) - \int_0^a \tilde{r}_t(\alpha) d\alpha$. Differentiating both sides with respect to $a$ yields a closed-form expression for the gradient of cost:

$$\nabla_a \tilde{c}_t = a \nabla_a \tilde{r}_t. \tag{2}$$

This equation allows us to directly obtain $\nabla_a \tilde{c}_t(a)$ using $\nabla_a \tilde{r}_t(a)$, avoiding separate gradient estimation for costs.

**Non-depletion Step: From Gradients on Actions to Policy Gradients.** To obtain the gradient with respect to the policy parameters $\theta$, we apply the chain rule:

$$\nabla_\theta \tilde{r}_t = \nabla_{a_t} \tilde{r}_t \nabla_\theta a_t, \quad \nabla_\theta \tilde{c}_t = \nabla_{a_t} \tilde{c}_t \nabla_\theta a_t.$$

We first consider the case of deterministic policies $\mu_\theta$, i.e., $a_t = \mu_\theta(s_t)$. We further expand

$$\nabla_\theta a_t = \nabla_\theta \mu_\theta(s_t) + \nabla_{s_t} \mu_\theta(s_t) \nabla_\theta s_t,$$

where the first term characterizes the direct influence of $\theta$ on $a_t$, the second term characterizes how the policy $\theta$ affects earlier actions, thereby affecting the current state $s_t$ and, consequently, $a_t$. For stochastic policies, we re-parameterize $\pi_\theta$ as $a_t = f_\theta(s_t; \epsilon_t)$, where $\epsilon_t \sim \mathcal{N}(0, 1)$ is an independent sample from a standard Gaussian distribution. Then we have a similar expression $\nabla_\theta a_t = \nabla_\theta f_\theta(s_t; \epsilon_t) + \nabla_{s_t} f_\theta(s_t; \epsilon_t) \nabla_\theta s_t$. According to Theorem 3.1, the state contains the remaining budget $B_t = B - \sum_{\tau=1}^{t-1} c_\tau$. Thus, the term $\nabla_\theta s_t$ depends on the gradients of all previous costs, $\nabla_\theta \tilde{c}_\tau$ for $\tau < t$, which is computed recursively.

**Depletion Step.** For a depletion step $u \in [T]$, the reward highly depends on the remaining budget $B_u$. Since the bidding process terminates immediately when the budget exhausts, the exact gradient $\nabla_\theta \tilde{r}_u$ depends on the order of arriving impressions in step $u$, which could be noisy. To simplify this, we approximate the reward as that achieved by the action $a_u^* = c_u^{-1}(B_u)$ that spends exactly $B_u$. The gradient is then approximated as follows:

$$\nabla_\theta \tilde{r}_u \approx \frac{\partial \tilde{r}_u}{\partial B_u} \cdot \nabla_\theta B_u = \frac{1}{a_u^*} \cdot \nabla_\theta B_u,$$

where the equality follows from Equation (2) and the identity $\partial r_u / \partial B_u = (\partial r_u / \partial a_u)/(\partial c_u / \partial a_u) = 1/a_u$ at $a_u = a_u^*$.

**The Final Gradient Estimator.** Combining both cases, the complete gradient estimator is

$$\nabla_\theta \tilde{J}(\pi_\theta; \mathcal{I}) = \begin{cases} \sum_{t=1}^{u-1} \nabla_\theta \tilde{r}_t + \frac{\nabla_\theta B_u}{c_u^{-1}(B_u)}, & \text{if the budget depletes in step } u \leq T, \\ \sum_{t=1}^{T} \nabla_\theta \tilde{r}_t, & \text{if budget remains until episode ends.} \end{cases} \tag{3}$$

This estimator is fully compatible with modern automatic differentiation frameworks (e.g., PyTorch (Paszke et al., 2019)). Implementation requires only custom gradient definitions for the per-step reward and cost functions with respect to actions and states.

Algorithm 1 describes the training procedure of FOB. Given a buffer $\mathcal{B}$ containing multiple historical instances, the algorithm simply performs stochastic gradient descent with our estimator Equation (3). This offline paradigm is standard in auto-bidding (He et al., 2021; Korenkevych et al., 2024) for safety and stability (Li et al., 2024). If online interactions are allowed, we deploy the current policy to collect new instances, and add them to the buffer. FOB is remarkably simple compared with standard deep RL approaches: it employs only an actor network to represent the policy—no critic, no temporal difference (TD) learning, no target networks, and no off-policy replay buffers. FOB is also flexible enough to extend to more general objectives and constraints, as discussed in Appendix D.

---

[3]By the generality of Myerson's Lemma, our result holds not only when the impressions are sold through second-price auctions, but for all single-parameter truthful auctions, e.g., the truthful multi-slot auction (Aggarwal et al., 2019). Our proof establishes this generalization.

---

**Algorithm 1** First-Order Policy Gradient for Auto-Bidding (FOB)

---

1: Initialize parameter vector $\theta$, instance buffer $\mathcal{B}$
2: **for** each learning episode **do**
3:     Sample a batch of $K$ instances $\{\mathcal{I}_1, \cdots, \mathcal{I}_K\}$ from $\mathcal{B}$
4:     Run policy $\pi_\theta$ on the $K$ instances
5:     Compute the gradient $\frac{1}{K}\sum_{k=1}^{K} \nabla_\theta \tilde{J}(\pi_\theta; \mathcal{I}_k)$ by Equation (3) and update the policy $\pi_\theta$ for one step with Adam
6:     **if** online interaction enabled **then**
7:         Deploy $\pi_\theta$, collect new instance $\mathcal{I}$ and add to buffer $\mathcal{B} \leftarrow \mathcal{B} \cup \{\mathcal{I}\}$
8:     **end if**
9: **end for**

---

## 5 EXPERIMENTS

We evaluate the effectiveness of FOB against standard zeroth-order RL methods and domain-specific auto-bidding baselines. Code is available at `https://anonymous.4open.science/r/FOB-979E`.

We use AuctionNet (Su et al., 2024), a publicly available benchmark derived from real-world ad auctions. Each episode contains approximately $10^5$ impressions and is divided into 48 steps. For each impression, we simulate a 48-player single-slot second-price auction. From the perspective of a single advertiser, this generates one instance. We collect 50 instances (5 million impressions total), splitting them into 30 training and 20 test instances. This setup mirrors real-world scenarios where an advertiser trains on historical bidding logs to optimize future performance, consistent with prior work (Wu et al., 2018; He et al., 2021; Wang et al., 2022).

We compare to classic RL algorithms: PPO (Schulman et al., 2017), an on-policy algorithm based on zeroth-order policy gradients; SAC (Haarnoja et al., 2018), an off-policy maximum entropy actor-critic algorithm; TD3 (Fujimoto et al., 2018), an off-policy algorithm based on deterministic policy gradients (Silver et al., 2014), improving upon DDPG (Lillicrap et al., 2015). We also include USCB (He et al., 2021), a state-of-the-art RL-based auto-bidding algorithm that modifies DDPG with a heuristic surrogate critic objective.

The policies are trained to handle multiple budget levels: $B \in \{150, 200, 250, 300, 350, 400\}$. During training, each sampled instance is assigned a random budget level. One epoch consists of 30 episodes (one per training instance). For FOB, we use a three-layer fully connected policy network (128–64–64) optimized with Adam (learning rate $5 \times 10^{-4}$). By default, reward gradients are approximated via piecewise-linear approximation. Baseline hyperparameters mainly follow Stable-Baselines3 defaults (Raffin et al., 2021); full details are in Appendix E. We report *normalized return* $(R/R^*)$, the standard metric in auto-bidding literature (He et al., 2021; Mou et al., 2022; Li et al., 2024), defined as the ratio of policy return to optimal return (Eq. 1). For each budget level, we compute the sum of returns over all 20 test instances, divided by the sum of optimal returns.

### 5.1 RESULTS

Table 1: Performance comparison on test instances. All methods are trained for 150 epochs ($\sim 2 \times 10^5$ steps) to convergence, except USCB (70 epochs to avoid overfitting). Results show mean $\pm$ std over 5 random seeds.

| Budget | FOB | PPO | TD3 | SAC | USCB |
|---|---|---|---|---|---|
| 150 | **0.819 $\pm$ 0.003** | 0.720 $\pm$ 0.055 | 0.752 $\pm$ 0.015 | 0.782 $\pm$ 0.008 | 0.793 $\pm$ 0.011 |
| 200 | **0.819 $\pm$ 0.003** | 0.734 $\pm$ 0.045 | 0.764 $\pm$ 0.016 | 0.793 $\pm$ 0.010 | 0.797 $\pm$ 0.009 |
| 250 | **0.819 $\pm$ 0.003** | 0.743 $\pm$ 0.036 | 0.771 $\pm$ 0.018 | 0.802 $\pm$ 0.009 | 0.801 $\pm$ 0.009 |
| 300 | **0.821 $\pm$ 0.003** | 0.748 $\pm$ 0.029 | 0.775 $\pm$ 0.021 | 0.808 $\pm$ 0.009 | 0.803 $\pm$ 0.008 |
| 350 | **0.822 $\pm$ 0.003** | 0.748 $\pm$ 0.024 | 0.781 $\pm$ 0.022 | 0.813 $\pm$ 0.009 | 0.805 $\pm$ 0.007 |
| 400 | **0.824 $\pm$ 0.003** | 0.744 $\pm$ 0.024 | 0.786 $\pm$ 0.022 | 0.816 $\pm$ 0.008 | 0.806 $\pm$ 0.007 |
| Training Time (s) | 10067 $\pm$ 66 | 14056 $\pm$ 41 | 18477 $\pm$ 95 | 31891 $\pm$ 79 | 16237 $\pm$ 60 |

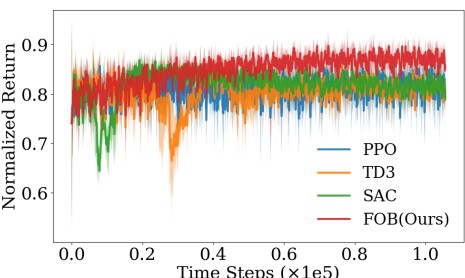

Figure 3: Learning curves comparison on training instances. Shades depict standard deviation over 5 seeds. Smoothed with WMA, $\alpha = 0.2$.

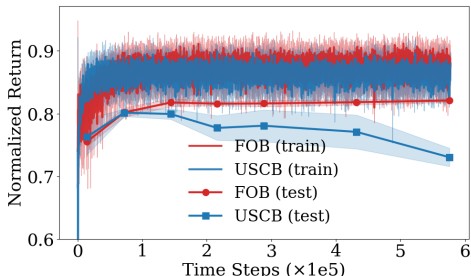

Figure 4: Training vs. test performance of FOB and USCB. Shaded areas depict standard deviation over 5 seeds.

**Comparison Against Baselines.** As shown in Figure 3, FOB exhibits superior training stability and convergence speed compared to zeroth-order RL algorithms (PPO, TD3, SAC). Due to the highly stochastic nature of the environment, the original training curves are fluctuating, so we apply strong smoothing (WMA with $\alpha = 0.2$). The plot without smoothing is available in Appendix F. Table 1 shows that, on the test set, FOB achieves the highest normalized return across all budget levels. Notably, baseline methods (TD3, SAC, USCB) struggle at lower budgets ($B \leq 250$), while FOB maintains strong performance. Table 1 also reports the training time of all methods for 150 epochs. Although the $T-$step backpropagation in FOB requires time, its overall training is still fastest among baselines due to the algorithmic simplicity. We further compare FOB with USCB, the most competitive baseline considering both performance and training time. USCB employs a heuristic surrogate objective for critic training, $Q^{\dagger}(s_t, a_t) = r(s_t, a_t) + \mathbb{E}[\sum_{\tau=t+1}^{T} r(s_\tau, a_t)]$, which fixes the action $a_t$ across future steps. While this avoids TD learning and stabilizes training, Figure 4 reveals it causes severe overfitting: test performance significantly degrades as training progresses. In contrast, FOB shows stable generalization to the test set.

**Algorithm Ablations.** We evaluate how gradient approximation choices affect FOB's performance. We compare the default setting, i.e., piecewise-linear approximation for rewards and the closed-form expression Equation (2) for costs, with the following variants: FOB-SG, using SG filter for rewards and Equation (2) for costs; FOB-SmoothC, using piecewise-linear approximation for both rewards and costs; FOB-SG-SmoothC, using SG filter for both rewards and costs. As shown in Table 2, performance differences across variants are marginal (all within 0.6%), confirming FOB's robustness to gradient approximation choices. Using closed-form cost gradients reduces training time by 3–8% compared to smoothing.

Table 2: Test-set performance comparison of FOB with various gradient approximation choices. Performances are averaged over different budgets. Full results across budgets are in Appendix F.

|  | FOB | FOB-SG | FOB-SmoothC | FOB-SG-SmoothC |
|---|---|---|---|---|
| Normalized Return | $0.820 \pm 0.003$ | $0.821 \pm 0.005$ | $0.819 \pm 0.006$ | $0.819 \pm 0.007$ |
| Training Time (s) | $10067 \pm 66$ | $11311 \pm 54$ | $10956 \pm 45$ | $11728 \pm 71$ |

## 6 CONCLUSION

In this work, we presented FOB, a novel policy gradient method for step-level auto-bidding that exploits the unique structural properties of real-world bidding environments. Unlike standard reinforcement learning approaches that treat auto-bidding in auctions as black-box MDPs, FOB leverages analytical gradients to directly optimize bidding policies. Future directions include: Extending FOB to non-truthful auctions, such as first-price auctions and generalized second-price (GSP) auctions; Extending FOB to deal with other advertiser objectives, as discussed in Appendix D.

# 7 ETHICS STATEMENT

The research is conducted with scientific rigor and transparency, using publicly available data that simulates real-world advertising auctions without exposing sensitive user information. Our method, FOB, is designed to improve the efficiency and stability of auto-bidding systems, which can lead to more effective and fairer ad allocation. The authors affirm that this work respects privacy, avoids harm, and upholds the principles of responsible AI research as outlined in the ICLR Code of Ethics.

# 8 REPRODUCIBILITY STATEMENT

All relevant code in our experiment, including the implementation of FOB as well as all baseline methods, simulation environments, and trained models, will be made available in the GitHub repository provided in the main paper. Detailed descriptions of the experimental setup, including hyperparameter settings can be found in Appendix E.

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

# A  THEORETICAL RESULTS AND PROOFS

## A.1  THE OPTIMAL BIDDING FORMULA

Consider a budget-constrained bidding problem with $n$ impressions, where each impression $i$ has a value $v_i > 0$ and a price $p_i > 0$. Let $B > 0$ denote the budget constraint. In the following, we explicitly give the form of optimal bidding parameter $a$, and prove its near-optimality.

**The optimal parameter $a$.**  Sort the impressions in non-decreasing order of price-to-value ratio $r_i = p_i/v_i$. We use the subscript $(i)$ denotes the $i$-th impression in this sorted order, so that $r_{(1)} \leq r_{(2)} \leq \cdots \leq r_{(n)}$. Define $k$ as the largest integer in $[n]$ such that $\sum_{i=1}^{k} p_{(i)} \leq B$ (with $\sum_{i=1}^{0} p_{(i)} = 0$). Consider the linear bidding strategy with parameter:

$$a = \begin{cases} 0 & \text{if } k = 0, \\ r_{(k+1)} & \text{if } 0 < k < n, \\ r_{(n)} + \epsilon & \text{if } k = n \text{ (for any } \epsilon > 0), \end{cases} \tag{4}$$

which wins impression $i$ if and only if $b_i = av_i > p_i$.

Intuitively, the parameter $a$ is a threshold that selects (wins) all impressions with $r_i < a$, and rejects (loses) all impressions with $r_i \geq a$. The optimal $a$ should select as many low price-to-value impressions as possible (i.e., respecting the budget constraint). The following theorem shows that, under mild assumptions, the difference between the optimal value and the value achieved by the linear bidding formula is not larger than the maximum value of any single impression. Since the impression number is typically huge in real-world applications, this loss is negligible.

**Theorem A.1.** *Let $V^*$ be the optimal value of the bidding problem in Equation* (1) *, $V_{bidding}$ be the total value achieved by the linear bidding formula with the bidding parameter given in Equation* (4). *Assuming for any two different impressions $i$ and $j$, we have $r_i \neq r_j$. That is, there does not exist two impressions with exactly the same price-to-value ratio. Then,*

$$V^* - V_{bidding} \leq \max_{1 \leq i \leq n} v_i.$$

*Proof.* The optimal value of the bidding problem in Equation (1) is equal to that of the following integer program:

$$V^* = \max_{\mathbf{x} \in \{0,1\}^n} \sum_{i=1}^n v_i x_i \quad \text{s.t.} \quad \sum_{i=1}^n p_i x_i \leq B.$$

Let $V_{\text{LP}}$ be the optimal value of the following linear relaxation of the integer program:

$$V_{\text{LP}} = \max_{\mathbf{x} \in [0,1]^n} \sum_{i=1}^n v_i x_i \quad \text{s.t.} \quad \sum_{i=1}^n p_i x_i \leq B.$$

Since the linear relaxation expands the decision space of the original integer program, we have $V_{\text{LP}} \geq V^*$. Thus, we only need to prove $V_{\text{LP}} - V_{\text{bidding}} \leq \max_{1 \leq i \leq n} v_i$.

Let $k$ be the largest integer such that $C_k = \sum_{i=1}^k p_{(i)} \leq B$. The residual budget is $B' = B - C_k \geq 0$. We analyze three cases:

Case 1: $k = 0$. The optimal solution of the LP only affords a $B/p_{(1)}$ fraction of the first impression, giving $V_{\text{LP}} = B \cdot v_{(1)}/p_{(1)}$. The bidding formula buys nothing, leading to $V_{\text{bidding}} = 0$. Since $B < p_{(1)}$, we have $V_{\text{LP}} - V_{\text{bidding}} = B \cdot v_{(1)}/p_{(1)} \leq v_{(1)} \leq \max_{1 \leq i \leq n} v_i$.

Case 2: $0 < k < n$. Set $a = r_{(k+1)}$. Since $r_{(1)} < \cdots < r_{(k)} < r_{(k+1)}$, the bidding formula wins impressions $i = 1, \ldots, k$, achieving $V_{\text{bidding}} = \sum_{i=1}^k v_{(i)}$. The LP relaxation solution is:

$$x_{(i)}^* = \begin{cases} 1 & \text{if } i \leq k, \\ B'/p_{(k+1)} & \text{if } i = k+1, \\ 0 & \text{if } i > k+1, \end{cases}$$

with value $V_{\text{LP}} = \sum_{i=1}^k v_{(i)} + \left(\frac{B'}{p_{(k+1)}}\right) v_{(k+1)}$. Since $0 \leq B'/p_{(k+1)} < 1$, we have

$$V_{\text{LP}} < \sum_{i=1}^k v_{(i)} + v_{(k+1)} = V_{\text{bidding}} + v_{(k+1)}.$$

which implies $V_{\text{LP}} - V_{\text{bidding}} < v_{(k+1)} \leq \max_i v_i$.

Case 3: $k = n$. The bidding formula set $a > r_{(n)}$, which wins all impressions. The LP solution also assigns $x_i = 1$ for all impressions. Thus, $V_{\text{bidding}} = \sum_{i=1}^n v_i = V_{\text{LP}}$.

Concluding the three cases proves the theorem. $\qquad\square$

## A.2  PROOF OF THE REDUCED STATE SPACE

*Proof of Theorem 3.1.* First, notice that in the three-dimensional state $s_t = (t, B, B_t)$, the time step $t$ is naturally contained in the subscript, the total budget $B$ is fixed across steps [4], thus only $B_t$ is useful for planning. We prove that $\phi(S_t) = B_t$ is a model-irrelevance abstraction (bisimulation) (Li et al., 2006; Givan et al., 2003), thereby establishing that the optimal policy in the new MDP with state $s_t = B_t$ remains optimal in the original MDP.

Let $\mathcal{S}$ denote the state space of the original MDP, and let $\mathcal{B} \subseteq \mathbb{R}$ be the state space of the new MDP. Consider any two states $S_t, S_t' \in \mathcal{S}$ such that $\phi(S_t) = \phi(S_t') = B_t$. According to Definition 3 in Li et al. (2006), we must verify two conditions:

1. Reward Equivalence: For any action $a_t \in \mathcal{A}$,

$$\mathbb{E}[r_t | S_t, a_t] = \mathbb{E}[r_t | S_t', a_t].$$

2. Transition Equivalence: For any action $a_t \in \mathcal{A}$ and any measurable set $X \subseteq \mathcal{B}$,

$$\Pr(B_{t+1} \in X \mid S_t, a_t) = \Pr(B_{t+1} \in X \mid S_t', a_t).$$

---

[4] $B$ is involved in the state for the purpose of training a unified policy for different budget sizes.

We first prove reward equivalence. The immediate reward $r_t$ is determined by the action $a_t$, the impression batch $\gamma_t$, and the remaining budget $B_t$. To see this, recall the generation process of $r_t$: the reward for non-depletion steps $r_t = \sum_{i=1}^{n_t} v_{t,i}\mathbb{I}\{a_t \cdot v_{t,i} > p_{t,i}\}$ only involve $a_t$ and $\gamma_t$, the criterion for depletion (i.e., $B_t < \sum_{i=1}^{n_t} p_{t,i}\mathbb{I}\{a_t \cdot v_{t,i} > p_{t,i}\}$) involves $a_t, \gamma_t$ and $B_t$, and the reward upon depletion also involves $a_t, \gamma_t$ and $B_t$. The randomness in $r_t$ only comes from $\gamma_t$. Since $\gamma_t \sim P_t$ is independent of the history, the conditional distribution of $r_t$ given $(B_t, a_t)$ is identical for all $S_t, S'_t$ satisfying $\phi(S_t) = \phi(S'_t) = B_t$. This implies that the conditional expectations, $\mathbb{E}[r_t|S_t, a_t]$ and $\mathbb{E}[r_t|S'_t, a_t]$, are identical.

We now prove transition equivalence. The next state is $B_{t+1} = B_t - c_t$, where $c_t$ is the cost incurred at step $t$. Similar to the reward, the cost $c_t$ is determined by $a_t, \gamma_t$, and $B_t$. As $\gamma_t \sim P_t$ is independent of history, the distribution of $c_t$ conditioned on $(B_t, a_t)$ is identical for all $S_t, S'_t$ with $\phi(S_t) = \phi(S'_t) = B_t$. Consequently, for any measurable set $X \subseteq \mathcal{B}$,

$$\Pr(B_{t+1} \in X \mid S_t, a_t) = \Pr(B_{t+1} \in X \mid B_t, a_t) = \Pr(B_{t+1} \in X \mid S'_t, a_t).$$

Since both conditions are satisfied, according to Li et al. (2006), $\phi$ is a model-irrelevance abstraction. By Theorem 3 in Li et al. (2006), the optimal policy in the new MDP with state $s_t = B_t$ is optimal in the original MDP. $\qquad\square$

### A.3 PROOF OF MYERSON'S LEMMA FOR GRADIENT COMPUTATION

We begin by introducing Myerson's Lemma (Myerson, 1981), the cornerstone of single-parameter mechanism design:

**Theorem A.2** (Myerson's lemma (Myerson, 1981)). *In a single-parameter auction, consider any specific player, and fixing any other players' bids. Let $x(b)$ be the allocation function with respect to the player's bid $b$, and $p(b)$ be the payment function. A mechanism is truthful if and only if: (i) the allocation $x(b)$ is monotone non-decreasing in the bid $b$, and, (ii) the payment is given by*

$$p(b) = b \cdot x(b) - \int_0^b x(z)dz.$$

This allows us to prove the following result, which generalizes Proposition 4.1.

**Proposition A.3.** *Consider the reward function $r_t(a) = \sum_{i=1}^{n_t} v_{t,i}x_{t,i}(a \cdot v_{t,i})$ and the cost function $c_t(a) = \sum_{i=1}^{n_t} p_{t,i}(a \cdot v_{t,i})$, where $x_{t,i}(\cdot)$ and $p_{t,i}(\cdot)$ are the allocation and payment functions of truthful mechanisms. The following identity holds, $c_t(a) = ar_t(a) - \int_0^a r_t(\alpha)d\alpha$.*

*Proof.* Fix any step $t \in [T]$ and impression $i \in [n_t]$. According to Myerson's lemma, we have

$$p_{t,i}(a \cdot v_{t,i}) = a \cdot v_{t,i} \cdot x_{t,i}(a \cdot v_{t,i}) - \int_0^{a \cdot v_{t,i}} x_{t,i}(z)dz.$$

for any $a$. Let $\alpha = z/v_{t,i}$, change variable in the integral

$$p_{t,i}(a \cdot v_{t,i}) = a \cdot v_{t,i} \cdot x_{t,i}(a \cdot v_{t,i}) - \int_0^a v_{t,i} \cdot x_{t,i}(\alpha \cdot v_{t,i})d\alpha.$$

Summing up over all impressions,

$$\sum_{i=1}^{n_t} p_{t,i}(a \cdot v_{t,i}) = \sum_{i=1}^{n_t} a \cdot v_{t,i} \cdot x_{t,i}(a \cdot v_{t,i}) - \sum_{i=1}^{n_t} \int_0^a v_{t,i} \cdot x_{t,i}(\alpha \cdot v_{t,i})d\alpha.$$

By linearity of integration, we can interchange the sum and integral in the last term:

$$\sum_{i=1}^{n_t} p_{t,i}(a \cdot v_{t,i}) = \sum_{i=1}^{n_t} a \cdot v_{t,i} \cdot x_{t,i}(a \cdot v_{t,i}) - \int_0^a \sum_{i=1}^{n_t} v_{t,i} \cdot x_{t,i}(\alpha \cdot v_{t,i})d\alpha.$$

Observe that by definition of $r_t(a)$ and $c_t(a)$, the above identity is equivalent to the desired identity

$$c_t(a) = ar_t(a) - \int_0^a r_t(\alpha)d\alpha.$$

$\qquad\square$

Proposition 4.1 is a special case of Proposition A.3, focusing on single-item second-price auctions. In this special case, the allocation and payment functions are step functions: $x_{t,i}(b) = \mathbb{I}\{b > p_{t,i}\}$ and $p_{t,i}(b) = p_{t,i}\mathbb{I}\{b > p_{t,i}\}$, which satisfy the requirement of Myerson's lemma. Then $r_t(a) = \sum_{i=1}^{n_t} v_{t,i}\mathbb{I}\{a \cdot v_{t,i} > p_{t,i}\}$ and $c_t(a) = \sum_{i=1}^{n_t} p_{t,i}\mathbb{I}\{a \cdot v_{t,i} > p_{t,i}\}$ are consistent with the main text.

## B  FURTHER RELATED WORK

**Budget-Constrained Auto-bidding.**   The problem of auto-bidding under budget constraints (Balseiro & Gur, 2019; Balseiro et al., 2023) is a special case of online resource allocation problems (Devanur et al., 2019; Roth, 2023; Balseiro et al., 2020). Theoretical works in this line primarily focus on designing algorithms with provable low-regret guarantees in the asymptotic regime—i.e., as the time horizon tends to infinity (Balseiro & Gur, 2019; Balseiro et al., 2020). In contrast, real-world auto-bidding systems (Ou et al., 2023) are different: campaigns run over a finite horizon (e.g., one day), the impression distribution is often almost known due to abundant historical data, and decisions are step-level instead of impression-level.

To address these practical considerations, recent applied literature has largely adopted the step-level auto-bidding framework formalized in Section 3. Early efforts employed heuristic rules to adjust the bidding parameter (Lee et al., 2013; Geyik et al., 2016). Subsequently, classical control techniques, such as PID controllers, were introduced (Yang et al., 2019; Karlsson, 2020; Zhang et al., 2022). Controllers are effective in smoothing budget expenditure but do not directly optimize rewards. This has motivated the adoption of reinforcement learning (Wu et al., 2018; He et al., 2021; Mou et al., 2022; Wang et al., 2022; Zhang et al., 2023; Li et al., 2024; Korenkevych et al., 2024; Guo et al., 2024). Most recent RL-based approaches focus on different aspects, including handling multiple constraints (He et al., 2021; Wang et al., 2022), enabling safe online exploration Mou et al. (2022); Li et al. (2024), incorporating personalization (Zhang et al., 2023), and handling non-Markov environments (Guo et al., 2024). In contrast, our work revisits a minimalist setting of the budget-constrained auto-bidding problem, in which we uncover and leverage structural properties that have been previously overlooked.

**First-Order Policy Gradient in RL.**   First-order (or pathwise) policy gradients offer a promising advantage over zeroth-order estimators by typically yielding lower variance (Mohamed et al., 2020). Their application in RL is primarily based on two approaches. The first involves constructing differentiable physics-based simulators (Xu et al., 2022; Xing et al., 2025) to give explicit gradients. While powerful in domains like robotics, this approach restricts to physics-based systems, demands substantial engineering effort (Freeman et al., 2021), and is prone to numerical instabilities such as gradient explosion due to environment stiffness or discontinuities (Suh et al., 2022). The second line of work learns a differentiable world model (e.g., via neural networks) and backpropagates through it (Heess et al., 2015; Clavera et al., 2020; Li et al., 2021). Although more broadly applicable, the errors in the learned dynamics propagate into gradient errors (Li et al., 2021), potentially misleading policy learning.

In contrast, our work suggests that, the auto-bidding problem naturally admits first-order policy gradients without requiring a differentiable simulator or a learned model. Only historical traffic instances are required, which are already present in most industrial advertising systems. Moreover, the bidding environment is free of the physical stiffness that challenge differentiable simulators (Suh et al., 2022). Our proposed estimator, FOB, thus unlocks the benefits of first-order RL at minimal cost and remains compatible with recent algorithmic advances in this paradigm (Xu et al., 2022; Xing et al., 2025).

**Hindsight learning for structured MDPs.**   Our work is related to previous studies that exploit domain-specific information structures for RL applications. Sinclair et al. (2023) observe that many operations research problems, e.g., cloud resource management, airline revenue management, have specific information structures that allow counterfactual policy evaluation. They propose a hindsight learning framework that employs offline planners during training. In a similar spirit, for the specific domain of inventory control, Madeka et al. (2022); Alvo et al. (2023) utilize first-order algorithms that share the same high-level idea with our approach. In their problem, the objective function is inherently differentiable due to the problem's continuous nature. In contrast, differentiability in auto-bidding is not immediate—it relies on the key observation on impression density

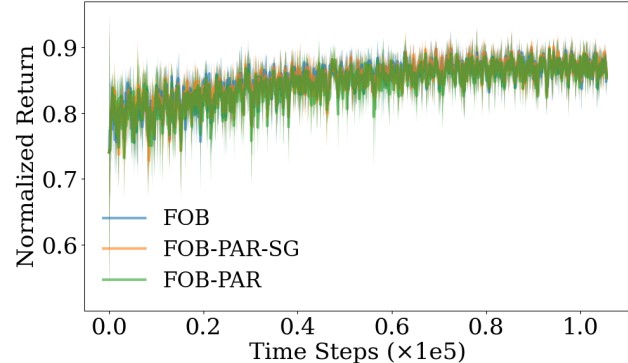

Figure 5: Learning curves under partial feedback. Mean $\pm$ std over 5 seeds, smoothed with weighted moving average ($\alpha = 0.2$). FOB maintains stable convergence despite limited price information.

(Observation 1). Moreover, FOB explicitly addresses early budget depletion, a challenge unique to budget-constrained bidding, and leverages auction-theoretic properties (Proposition 4.1) for gradient computation.

## C  HANDLING PARTIAL FEEDBACK

Throughout the main text, we assume that the platform reveals the clearing prices of all impressions (both won and lost), which is common in industrial practice when platforms train agents on behalf of advertisers (Decarolis et al., 2020; Wen et al., 2022). However, in some settings, only the prices of won impressions are observable—a scenario known as partial feedback. Here, we show that FOB naturally adapts to this setting by leveraging *one-sided gradient estimates*.

Under partial feedback, the agent observes only a subset of the full instance $\mathcal{I}$: for each step $t$, it sees $(v_{t,i}, p_{t,i})$ exclusively for impressions it wins, i.e., those satisfying $a_t > p_{t,i}/v_{t,i}$. These correspond to breakpoints to the left of the current action $a_t$. Crucially, this is sufficient to estimate the *left-hand derivative* of $r_t(\cdot)$ and $c_t(\cdot)$, enabling FOB to proceed with minor modifications:

1. *One-sided Piecewise-Linear Approximation:* Let $k$ be the number of winning impressions at step $t$, and let $\{a_{(1)} \leq \cdots \leq a_{(k)}\}$ denote the sorted ratios $p_{t,i}/v_{t,i}$. The gradient is approximated using the last segment:

$$\nabla_{a_t} \tilde{r}_t(a_t) = \frac{r_t(a_{(k)}) - r_t(a_{(k-1)})}{a_{(k)} - a_{(k-1)}}.$$

2. *One-sided Savitzky–Golay Filter:* We fit a quadratic polynomial to the $m$ winning impressions whose ratios are closest to (and less than) $a_t$. The derivative of the fitted curve at $a_t$ yields $\nabla_{a_t} \tilde{r}_t(a_t)$.

With these one-sided estimators, FOB can be trained online: deploy policy $\pi_\theta$ for one episode, collect the partial instance $\mathcal{I}_{\text{par}}$, compute the one-sided gradient, and perform a single policy update before the next deployment.

We evaluate this variant under the same experimental setup as Section 5. Results in Table 3 show that FOB-PAR (piecewise-linear) achieves performance nearly matching full-feedback FOB across all budgets, with a maximum drop of only 0.002 in normalized return. The SG-based variant (FOB-PAR-SG) performs slightly worse, likely due to over-smoothing with limited left-side points. Figure 5 further confirms that training remains stable and convergent under partial feedback. These findings demonstrate that FOB is robust to partial observability.

Table 3: Performance of FOB under partial feedback (mean $\pm$ std, 5 seeds). FOB-PAR uses one-sided piecewise-linear gradients; FOB-PAR-SG uses one-sided SG filtering.

| Budget | FOB (full) | FOB-PAR | FOB-PAR-SG |
|--------|-----------|---------|------------|
| 150 | $0.819 \pm 0.003$ | $0.817 \pm 0.010$ | $0.815 \pm 0.003$ |
| 200 | $0.819 \pm 0.003$ | $0.819 \pm 0.010$ | $0.814 \pm 0.004$ |
| 250 | $0.819 \pm 0.003$ | $0.820 \pm 0.009$ | $0.814 \pm 0.005$ |
| 300 | $0.821 \pm 0.003$ | $0.821 \pm 0.010$ | $0.815 \pm 0.005$ |
| 350 | $0.822 \pm 0.003$ | $0.822 \pm 0.009$ | $0.817 \pm 0.006$ |
| 400 | $0.824 \pm 0.003$ | $0.823 \pm 0.010$ | $0.818 \pm 0.006$ |

## D  HANDLING MORE GENERAL OBJECTIVES AND CONSTRAINTS

While our primary focus has been on maximizing total impression value under a budget constraint, FOB is inherently flexible and extends naturally to a broad class of auto-bidding objectives and constraints.

In this section, we discuss how to extend FOB to more general objectives via a prominent example, the Return-on-Average-Spend (ROAS) constraint (Feng et al., 2023), which requires that total cost does not exceed a fixed multiple of total reward:

$$\sum_{t=1}^{T} c_t \leq C_{\text{tgt}} \cdot \sum_{t=1}^{T} r_t,$$

where $C_{\text{tgt}} > 0$ is a target ROAS threshold. This constraint is widely adopted in practice (Yang et al., 2019; He et al., 2021; Wang et al., 2022; Feng et al., 2023) to align bidding with advertiser profitability goals.

**FOB in the Primal-Dual Framework.**  When the ROAS constraint is enforced in expectation, the problem becomes a Constrained Markov Decision Process (CMDP):

$$\max_{\theta} \; \mathbb{E}_{\mathcal{I} \sim P} \left[ \sum_{t=1}^{T} r_t \right] \quad \text{subject to} \quad \mathbb{E}_{\mathcal{I} \sim P} \left[ \sum_{t=1}^{T} c_t - C_{\text{tgt}} \cdot \sum_{t=1}^{T} r_t \right] \leq 0. \tag{5}$$

This is commonly addressed via primal-dual methods (Achiam et al., 2017; Tessler et al., 2019), which introduce a non-negative dual variable $\lambda \geq 0$ and reformulate Equation (5) as the saddle-point problem:

$$\min_{\lambda \geq 0} \max_{\theta} \; \mathbb{E}_{\mathcal{I} \sim P} \left[ \sum_{t=1}^{T} r_t - \lambda \left( \sum_{t=1}^{T} c_t - C_{\text{tgt}} \cdot \sum_{t=1}^{T} r_t \right) \right]. \tag{6}$$

Standard constrained RL algorithms solve this via a two-timescale optimization: an outer loop updates $\lambda$ using (sub)gradient descent on the constraint violation, while an inner loop optimizes the policy $\pi_\theta$ using a standard RL method with modified per-step rewards $r_t - \lambda(c_t - C_{\text{tgt}} r_t)$. FOB seamlessly integrates into this framework: its first-order gradient estimator can be applied directly to the inner-loop policy optimization.

**FOB for Non-additive Objectives.**  Beyond primal-dual approaches, one may also embed the constraint directly into the objective (often called scalarization in multi-objective RL (Roijers et al., 2013)) by optimizing a non-linear objective function of total reward and cost:

$$\max_{\theta} \; \mathbb{E}_{\mathcal{I} \sim P} \left[ g \left( \sum_{t=1}^{T} r_t, \sum_{t=1}^{T} c_t \right) \right],$$

where $g : \mathbb{R}^2 \to \mathbb{R}$ encodes the advertiser's preference. For instance, He et al. (2021) proposes

$$g(R, C) = R - \beta^{(C - C_{\text{tgt}} \cdot R)^+},$$

where $\beta > 0$ is a hyperparameter controlling penalty strength, and $x^+ = \max(x, 0)$. Such objectives are non-additive: They cannot be decomposed into per-step rewards, and thus fall outside the scope

of conventional RL algorithms that rely on Bellman equations. A common workaround is assigning zero reward at all intermediate steps and $g(R, C)$ only at the final step. This results in sparse, delayed rewards, which severely hampers policy learning.

FOB, however, handles these non-additive objectives naturally. Because it operates on full historical instances $\mathcal{I}$ and leverages the differentiability of the return with respect to policy parameters, FOB bypasses the credit assignment problem entirely. As long as the composite objective $g(\sum_t r_t, \sum_t c_t)$ is differentiable in the policy parameters $\theta$, which holds under mild smoothness conditions on $g$ and our surrogate reward/cost functions, FOB can compute first-order gradients via the chain rule. This makes it uniquely suited for auto-bidding problems with complex objectives.

# E EXPERIMENT DETAILS

All experiments are run on the same CPU (Intel(R) Xeon(R) Silver 4210R CPU @ 2.40GHz) and GPU (NVIDIA GeForce RTX 3090). All algorithms use fully connected neural networks with SiLU (Swish) activation functions and operate on a 3-dimensional state input $(t, B_t, B)$ and a 1-dimensional action output (bidding parameter). The actor networks across methods share a common backbone architecture of (128,64) hidden units. For all algorithms, a tanh function is applied to squash the sampled actions to $[-1, 1]$, then linearly scaled to $[50, 120]$. Below are the algorithm-specific configurations:

**FOB.** An actor network for a Gaussian policy. The network consists of two hidden layers (128,64), followed by two separate one-layer heads (64,1) for mean $\mu$ and log-standard deviation $\log \sigma$. During training, actions are sampled via reparameterization. There are no target actors, and no critic networks.

**PPO.** The same actor architecture as FOB. Also includes a separate critic network (V-function) with architecture (64,64,1), with a target critic using the same architecture.

**TD3.** A deterministic actor with architecture (128, 64, 1), with a target actor using the same architecture. TD3 also uses twin Q-networks, each (64,64,1), which take concatenated state-action inputs to mitigate overestimation bias. Both Q networks have target networks.

**SAC.** The same actor architecture as FOB and PPO. Like TD3, it employs twin Q-networks (64,64,1). Both Q networks have target networks.

**USCB.** A deterministic actor with the same architecture (128, 64, 1) as TD3. Its critic is a single Q-network (64,64,1) with a target network, aligning with its DDPG-based foundation.

The SG filter for FOB is implemented as follows: Sort the impressions by $r_i = p_{t,i}/v_{t,i}$ in ascending order, yielding sequences $\{c_{(j)}\}$, $\{v_{(j)}\}$, and $\{p_{(j)}\}$. Select impressions whose $r_i$ lies within a bandwidth $h > 0$ of $a$. Compute cumulative reward over the sorted local impressions: $R_{\text{cum}}^{(k)} = \sum_{j=1}^{k} v_{(j)}$. Fit a polynomial with degree $d$ to the cumulative reward via least squares. The estimated derivative at $a$ corresponds to the linear coefficient of the fitted polynomial. In our experiments, we use $h = 3$ and $d = 2$ (quadratic fit) by default.

# F ADDITIONAL RESULTS

We include additional results in this section.

Figure 6 presents the original training curves without smoothing. The curves fluctuate due to highly stochastic environments.

Table 5 provides full results of our ablations on FOB. The performance differences across variant gradient approximation schemes are marginal.

Table 4: Algorithm-specific hyperparameters used in experiments.

| Hyperparameter | FOB | PPO | TD3 | SAC | USCB |
|---|---|---|---|---|---|
| Actor learning rate | $5 \times 10^{-4}$ | $1 \times 10^{-4}$ | $1 \times 10^{-3}$ | $3 \times 10^{-4}$ | $3 \times 10^{-4}$ |
| Critic learning rate | — | $1 \times 10^{-4}$ | $1 \times 10^{-3}$ | $3 \times 10^{-4}$ | $1 \times 10^{-3}$ |
| Max gradient norm | 0.5 | 0.5 | 0.5 | 0.5 | 0.5 |
| Discount factor $\gamma$ | 1.0 | 1.0 | 1.0 | 1.0 | 1.0 |
| GAE $\lambda$ | — | 0.95 | — | — | — |
| Mini-epochs | 1 | 10 | 1 | 1 | 1 |
| Policy type | Stochastic | Stochastic | Deterministic | Stochastic | Deterministic |
| Actor $\sigma(s)$ | yes | yes | no | yes | no |
| Actor $\log(\sigma)$ | [-10,2] | [-10,2] | — | [-10,2] | — |
| Policy noise | — | — | 0.05 | — | 0.05 |
| Batch size | — | 64 | 256 | 256 | 32 |
| Num of Critics | — | 1 | 2 | 2 | 1 |
| Target update rate $\tau$ | — | 0.005 | 0.005 | 0.005 | — |
| Learning starts | — | — | 20000 | 5000 | — |
| Replay buffer | — | — | 50000 | 50000 | 50000 |
| PPO clip ratio $\epsilon$ | — | 0.2 | — | — | — |
| TD3 target policy noise | — | — | 0.2 | — | — |
| TD3 target policy clip | — | — | 0.5 | — | — |
| TD3 policy update delay | — | — | 2 | — | — |
| SAC Initial $\alpha$ | — | — | — | $\log(0.01)$ | — |
| SAC $\log(\alpha)$ learning rate | — | — | — | 1e-3 | — |
| SAC Target entropy | — | — | — | -1 | — |

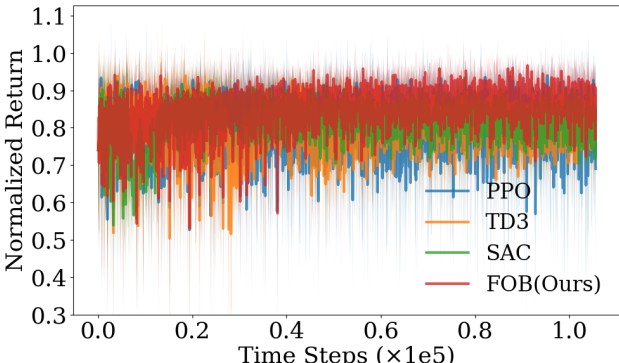

Figure 6: Original Learning curves training instances, using the same data as Figure 3, but without smoothing (WMA $\alpha = 1$).

## G  LLM USAGE

LLMs are used for improving the writing of the paper, and for writing the visualization codes.

Table 5: Performance comparison of FOB variants on test instances. Full version of Table 2. Results show mean $\pm$ std over 5 random seeds.

| Budget | FOB | FOB-SG | FOB-SmoothC | FOB-SG-SmoothC |
|---|---|---|---|---|
| 150 | $0.819 \pm 0.003$ | $0.816 \pm 0.006$ | $0.816 \pm 0.007$ | $0.814 \pm 0.009$ |
| 200 | $0.819 \pm 0.003$ | $0.817 \pm 0.006$ | $0.816 \pm 0.006$ | $0.815 \pm 0.008$ |
| 250 | $0.819 \pm 0.003$ | $0.819 \pm 0.006$ | $0.818 \pm 0.006$ | $0.816 \pm 0.007$ |
| 300 | $0.821 \pm 0.003$ | $0.821 \pm 0.005$ | $0.819 \pm 0.005$ | $0.818 \pm 0.006$ |
| 350 | $0.822 \pm 0.003$ | $0.823 \pm 0.005$ | $0.821 \pm 0.005$ | $0.820 \pm 0.005$ |
| 400 | $0.824 \pm 0.003$ | $0.826 \pm 0.004$ | $0.823 \pm 0.005$ | $0.822 \pm 0.004$ |

