# OpenReview forum: "Back Propagation through Auctions: First-Order Policy Gradient for Auto-Bidding"
_ICLR.cc/2026/Conference — Submitted to ICLR 2026_

### Official Review · Reviewer_UFLL · 2025-10-14

**Soundness:** 3
**Presentation:** 3
**Contribution:** 3
**Rating:** 4
**Confidence:** 2

**Summary:**

This paper studies the problem of automated bidding at the step level in fast-paced online ad auctions. The central insight is that the auto-bidding environments are different from typical MDPs in standard reinforcement learning problem settings, as they provide detailed feedback for each impression (i.e. values and clearing prices for individual auctions). Also, since there are so many impressions packed into each time step, the relationship between the bidding parameter and the step's reward/cost is more smooth. To resolve these problems, the authors develop ``FOB'', a first-order policy-gradient method that backpropagates through actual historical auction instances. At each step, they smooth out the piecewise-constant reward function $r_t(a)=\sum_i v_{t,i}\mathbf{1}{a,v{t,i}>p_{t,i}}$ into a differentiable version $\tilde r_t(a)$ and calculate $\nabla_a \tilde r_t(a)$ using either piecewise-linear slopes between breakpoints or a Savitzky–Golay fit. For the step cost $c_t(a)$, they apply Myerson's lemma to enforce $c_t(a)=a,r_t(a)-\int_0^a r_t(\alpha),d\alpha$, which gives them the analytical gradient $\nabla_a \tilde c_t(a)=a,\nabla_a \tilde r_t(a)$. The policy gradient across multiple steps then comes from the chain rule, with earlier actions affecting later ones through the remaining budget $B_t$. They treat budget depletion steps specially by differentiating with respect to $B_u$ using $\partial r_u/\partial B_u=1/a_u^\star$ where $a_u^\star=c_u^{-1}(B_u)$. On the theory side, they offer: (i) a reduced state abstraction result (Theorem 3.1) proving that $s_t=(t,B_t,B)$ is sufficient when step arrivals are independent, (ii) a derivation through Myerson establishing the identity they use for cost gradients, and (iii) a generalization showing the cost/reward linkage works for any single-parameter truthful mechanism. In experiments on AuctionNet (48 steps per episode; roughly $10^5$ impressions per episode), FOB beats PPO, TD3, SAC, and a specialized DDPG variant (USCB) in normalized return across budgets $B$ from 150 to 400, with quicker and more stable training. Ablation studies reveal comparable performance with different smoothing methods, and a partial-feedback variant (observing only prices of won impressions) using one-sided gradients performs nearly as well as full-feedback FOB. The code is promised, and the paper explicitly declares LLM usage for writing and visualization code.

**Strengths:**

(1) Clear identification of domain structure (counterfactual replay, high impression density) enabling first-order gradients.

(2) Succinct algorithm (actor-only; no TD/critics); stable training; competitive wall-clock.

(3) Solid empirical showing on AuctionNet; ablations on smoothing and partial feedback variant.

**Weaknesses:**

(1) Assumption on feedback completeness and realism: The approach assumes access to the clearing prices $p_{t,i}$ for losing auctions to construct the breakpoint set for $r_t(a)$ (and exact offline-optimal normalization). While the paper does discuss partial feedback and provides one-sided estimators, major platforms do not always release losing prices to advertisers; even if the platform itself trains on behalf of advertisers, this assumption should be delineated more precisely. Please quantify how performance degrades as the fraction of missing losing prices increases beyond the one-sided case (e.g., random hiding or biased hiding w.r.t. $p/v$), and whether calibration tricks can recover gradients.

(2) Smoothing bias and sensitivity: The surrogate $,\tilde r_t(a)$ is a smoothed version of a step function. The paper provides two smoothing methods (piecewise-linear and SG), but there is no analysis of the bias introduced by smoothing choices and hyper-parameters (e.g., SG bandwidth $h$, polynomial degree $d$, number of points, etc). How sensitive is FOB to these choices across different traffic regimes (e.g., fewer impressions per step, heavy-tailed $v$ or $p$)? An ablation across impression densities (e.g., subsampling auctions per step to simulate sparse steps) would help verify the “high-density” assumption’s necessity.

(3) Generality beyond second-price truthful auctions: The Myerson identity and the clean gradient link hinge on single-parameter truthful mechanisms. The paper mentions extensions, but first-price and GSP do not directly admit the same identity. The method would then need explicit smoothing for cost (and potentially different chain rules). It would be good to either include a small experiment or a derivation sketch for first-price auctions showing how FOB adapts and whether the observed stability persists.

Of course, it is impossible for me to complete a very in-depth inspection on the soundness or technical details on a paper whose topic is out of my personal research domain, due to the 2-week reviewing time window versus 5 assigned submissions most of which are with extensive derivations of equations. I just reported what I found and what I thought.

**Questions:**

Your one-sided estimator uses only winning breakpoints. How does FOB behave when the policy changes significantly between epochs (e.g., shifts $a_t$ upward) and the relevant left-neighborhood has very few wins? Do you employ any exploration or smoothing safeguards to avoid vanishing gradients in such cases?

---

> ### Author Response · Authors · 2025-11-22
> **Rebuttal to Reviewer UFLL**
>
> We sincerely thank the reviewer for the thoughtful and detailed evaluation of our paper. We appreciate your positive comments on our identification of the domain structure, the simplicity of the algorithm, and the empirical results. Below we respond to each weakness and the question.
>
> **(1) Feedback completeness and realism; performance with missing losing prices**
>
> As shown in Appendix C of the main submission, the performance gap between **full feedback** and the **one-sided partial-feedback** setting is extremely small (below 0.3% in normalized return). This already indicates that the effectiveness of FOB **does not rely critically on the full‑feedback assumption**. To address your request about intermediate regimes between full and one‑sided feedback, we ran additional experiments with controlled missing fractions of losing prices.
>
> We consider four variants:
>
> - **FOB (full)**: feedback on both sides (all winning and losing prices observed).
> - **FOB (PAR)**: one‑sided estimator with only left‑side (winning) feedback.
> - **FOB (50% PAR)**: full left‑side (winning) feedback; on the right side (losing auctions), 50% of breakpoints are randomly hidden.
> - **FOB (30% PAR)**: full left‑side feedback; on the right side, 70% of breakpoints are hidden (only 30% kept).
>
> The normalized returns across budgets are:
>
> | Budget | FOB (full)    | FOB-PAR       | FOB (50% PAR) | FOB (30% PAR) |
> | ------ | ------------- | ------------- | ------------- | ------------- |
> | 150    | 0.819 ± 0.003 | 0.817 ± 0.010 | 0.816 ± 0.007 | 0.819 ± 0.009 |
> | 200    | 0.819 ± 0.003 | 0.819 ± 0.010 | 0.818 ± 0.007 | 0.818 ± 0.007 |
> | 250    | 0.819 ± 0.003 | 0.820 ± 0.009 | 0.820 ± 0.005 | 0.820 ± 0.008 |
> | 300    | 0.821 ± 0.003 | 0.821 ± 0.010 | 0.821 ± 0.006 | 0.821 ± 0.006 |
> | 350    | 0.822 ± 0.003 | 0.822 ± 0.009 | 0.823 ± 0.006 | 0.824 ± 0.007 |
> | 400    | 0.824 ± 0.003 | 0.823 ± 0.010 | 0.824 ± 0.006 | 0.824 ± 0.008 |
>
> The differences among these four settings are almost negligible, showing that FOB remains effective over a wide range of feedback completeness levels.

---

> ### Author Response · Authors · 2025-11-22
> **Rebuttal to Reviewer UFLL (Cont'd)**
>
> **(2) Smoothing bias and sensitivity; impression-density ablations**
>
> **Smoothing choices.** In Table 2 of the main paper, we already include an ablation comparing piecewise‑linear smoothing and a Savitzky–Golay (SG) filter with parameters $(h=3,d=2)$, showing that both work well. To further address your concern, we ran a broader sweep over SG parameters: window half‑width $h \in \{3,5,7\}$ and polynomial degree $d \in \{2,3\}$.
>
> The results are:
>
> | Budget | FOB-SG (h=3, d=2) | FOB-SG (h=3, d=3) | FOB-SG (h=5, d=2) | FOB-SG (h=5, d=3) | FOB-SG (h=7, d=2) | FOB-SG (h=7, d=3) |
> | ------ | ----------------- | ----------------- | ----------------- | ----------------- | ----------------- | ----------------- |
> | 150    | 0.816±0.006       | 0.816±0.002       | 0.815±0.004       | 0.816±0.003       | 0.821±0.008       | 0.816±0.002       |
> | 200    | 0.817±0.006       | 0.816±0.001       | 0.817±0.004       | 0.817±0.003       | 0.823±0.007       | 0.817±0.002       |
> | 250    | 0.819±0.006       | 0.817±0.001       | 0.818±0.004       | 0.819±0.002       | 0.825±0.007       | 0.819±0.002       |
> | 300    | 0.821±0.005       | 0.819±0.001       | 0.820±0.004       | 0.822±0.002       | 0.827±0.006       | 0.821±0.002       |
> | 350    | 0.823±0.005       | 0.821±0.001       | 0.822±0.003       | 0.824±0.001       | 0.829±0.005       | 0.824±0.001       |
> | 400    | 0.826±0.004       | 0.823±0.002       | 0.824±0.003       | 0.827±0.001       | 0.832±0.005       | 0.827±0.001       |
>
> All these variants perform closely to one another, indicating that FOB is empirically robust to reasonable choices of SG window width and polynomial degree. Notably, degree $d=3$ typically produces lower variance than $d=2$.
>
> **Impression-density ablation.** To directly test the necessity and robustness of the high‑density assumption, we simulate environments with different impression densities. Concretely, we create sparser versions by merging two impressions into one (summing their values and prices), and so on. This preserves the step‑wise reward and cost for a given action, while reducing the number of auctions per step and thus local smoothness.
>
> The following table reports normalized returns for FOB (using the piecewise‑linear estimator) at different effective densities versus SAC (the strongest RL baseline): “100%” corresponds to about $5 \times 10^5$ impressions per instance, aligned with raw AuctionNet data. “20% sparse (original)” corresponds to our original experimental setup with about $10^5$ impressions per episode, already much sparser than the raw AuctionNet data.
>
> | Budget | FOB (100%)  | FOB (80% sparse) | FOB (40% sparse) | FOB (20% sparse; original) | FOB (16% sparse) | FOB (10% sparse) | SAC         |
> | ------ | ----------- | ---------------- | ---------------- | -------------------------- | ---------------- | ---------------- | ----------- |
> | 150    | 0.825±0.004 | 0.827±0.003      | 0.823±0.005      | 0.819±0.003                | 0.772±0.005      | 0.643±0.005      | 0.782±0.008 |
> | 200    | 0.827±0.002 | 0.827±0.002      | 0.824±0.003      | 0.819±0.003                | 0.770±0.003      | 0.646±0.005      | 0.793±0.010 |
> | 250    | 0.828±0.002 | 0.829±0.002      | 0.828±0.002      | 0.819±0.003                | 0.770±0.004      | 0.650±0.005      | 0.802±0.009 |
> | 300    | 0.830±0.001 | 0.830±0.001      | 0.832±0.002      | 0.821±0.003                | 0.772±0.003      | 0.654±0.003      | 0.808±0.009 |
> | 350    | 0.831±0.002 | 0.832±0.001      | 0.836±0.001      | 0.822±0.003                | 0.773±0.002      | 0.658±0.003      | 0.813±0.009 |
> | 400    | 0.833±0.001 | 0.834±0.001      | 0.840±0.001      | 0.824±0.003                | 0.775±0.002      | 0.663±0.003      | 0.816±0.008 |
>
> These results show that: In the range from 100% to 20% density, FOB consistently outperforms SAC across all budgets. As density becomes extremely low (e.g., 10–16%), FOB’s performance gradually degrades.
>
> We note that, for computational efficiency, the main paper already uses about $10^5$ impressions per episode—roughly 20% of the full AuctionNet density of $5 \times 10^5$ impressions per instance—so our original experiments are already in a moderately sparse regime.

---

> ### Author Response · Authors · 2025-11-22
> **Rebuttal to Reviewer UFLL (Cont'd)**
>
> **(3) Generality beyond second‑price truthful auctions**
>
> In non‑truthful auctions (e.g., first‑price, GSP), there are in fact two changes compared with our setting: (i) Myerson’s lemma no longer holds, and (ii) the truly optimal bidding rule is generally **not** linear in value, so our linear form $b_{t,i} = a_t v_{t,i}$ is no longer theoretically optimal. We outline two ways FOB can still be applied.
>
> * **Simple practical adaptation: keep linear bidding.** One pragmatic approach is to keep the linear bidding rule $b_{t,i} = a_t v_{t,i}$ even under a first‑price auction (as in the case studied by [1]). In that case, the step‑level reward and cost become $
> r_t(a_t) = \sum_{i=1}^{n_t} v_{t,i}\mathbb{I}\{a_t v_{t,i} > p_{t,i}\}, \quad
> c_t(a_t) = \sum_{i=1}^{n_t} a_t v_{t,i}\mathbb{I}\{a_t v_{t,i} > p_{t,i}\}.$
> Myerson’s lemma is no longer applicable, but the form of the reward $r_t(a_t)$ is unchanged: it is still a non‑decreasing piecewise‑constant function of $a_t$, so our existing piecewise‑linear or Savitzky–Golay smoothing applies directly. Moreover, the cost $c_t(a_t)$ becomes a **continuous** (indeed, piecewise‑linear) function of $a_t$; we can therefore obtain directly a subgradient. In other words, **the main idea of FOB** (i.e., smoothing step‑level reward and cost and backpropagating through steps) **carries over** with minimal changes.
>
> * **More theoretically grounded extension: bid shading.** A more theoretically sound extension for non‑IC auctions is to incorporate bid shading explicitly using best‑response functions, following, for example, Lemma 1 in [2]. In this approach, we bid
> $
> b_{t,i} = \sigma_{t,i}(a_t v_{t,i}),
> $
> where $\sigma_{t,i}$ is the best‑response (shading) function for auction $(t,i)$. At the modeling level we can treat $\sigma_{t,i}$ as known; in practice it can be implemented as a learned, impression‑level parametric bid‑shading function based on features (see [3] on how to obtain $\sigma_{t,i}$). Then the step‑level reward and cost become
> $
> r_t(a_t) = \sum_{i=1}^{n_t} v_{t,i}\mathbb{I} \{\sigma_{t,i}(a_t v_{t,i}) > p_{t,i} \},
> \quad
> c_t(a_t) = \sum_{i=1}^{n_t} \sigma_{t,i}(a_t v_{t,i}) \mathbb{I}\{\sigma_{t,i}(a_t v_{t,i}) > p_{t,i}\}.
> $
> Here, $r_t(a_t)$ is still a piecewise‑constant function of $a_t$. Thus we can again apply piecewise‑linear or SG smoothing. Meanwhile, because $\sigma_{t,i}(\cdot)$ is continuous in its argument, $c_t(a_t)$ becomes a continuous function of $a_t$, and its gradient or subgradient can be computed directly via the chain rule.
>
> In summary, FOB’s core idea remains applicable when moving beyond truthful auctions.
>
>
>
> **Question: one-sided estimator behavior when policy shifts and wins are sparse on the left**
>
> The reviewer asked how FOB behaves under the one‑sided estimator when the policy changes substantially between epochs (e.g., shifting $a_t$ upward) so that the left neighborhood of $a_t$ has very few winning impressions, and whether we use any special exploration or smoothing safeguards to avoid vanishing gradients.
>
> In auto‑bidding, impression arrivals around any reasonable $a_t$ are very dense (see Figure 2 in the paper): for each step there are typically many impressions with prices both just below and just above the threshold $a_t v_{t,i} = p_{t,i}$. This makes it unlikely that the immediate left neighborhood of $a_t$ would contain very few wins in practice. Even if the count of left‑neighborhood wins is relatively small, this does not automatically imply that the estimated derivative is near zero: the slope between the last two winning breakpoints can still be significant, so vanishing gradients are not guaranteed. Empirically, our partial‑feedback experiments (including FOB‑PAR and the missing‑fraction variants above) achieve performance very close to full‑feedback FOB and show stable training behavior, without any special exploration mechanisms beyond the standard stochastic policy.
>
>
> [1] Conitzer et al., Pacing Equilibrium in First Price Auction Markets. Management Science 2021.
>
> [2] Susan et al., Multi‑Platform Budget Management in Ad Markets with Non‑IC Auctions.
>
> [3] Gligorijevic et al., Bid Shading in The Brave New World of First‑Price Auctions. CIKM 2020.

---

> > ### Comment · Reviewer_UFLL · 2025-11-22
> > **Thanks for your response**
> >
> > I appreciate the authors' efforts in responding to my concerns and my question in my review. For Weakness (1), I am totally convinced with the added experiments that have shown the robustness of FOB over full-info feedback and one-sided feedback. For Weakness (2), thanks you for pointing to Table 2 which I almost missed previously. After took a carefully look into the comparisons there (and those added in the rebuttal), I agree that the biasedness does not involve much in the performance. Also, since the performance significantly decreases only when density <16%, I agree that the high-density assumption is necessary. For my (only) question, it is reasonable as the overall empirical performance has justified the stability of gradient. While I am still curious about the mechanism behind (as it is not symmetry), I am convinced by evidences.
> >
> > Since I am not an expert in online auction, I felt like I could not totally understand your explanation on Weakness (3), but it sounds plausible as the high-level idea works in the potential generalization.
> >
> > Overall, this is a good work, and the rebuttal is exhaustive. I have increased my score and confidence accordingly.

---

> > > ### Author Response · Authors · 2025-11-25
> > >
> > > We sincerely thank the reviewer for the updated assessment and increased score and confidence. Your questions helped us strengthen the paper’s exposition and empirical support.

---

### Official Review · Reviewer_khCL · 2025-10-27

**Soundness:** 2
**Presentation:** 3
**Contribution:** 2
**Rating:** 2
**Confidence:** 4

**Summary:**

Assuming that the impression sequence is independent of the bidding actions, the paper proposes a auto-bidding optimization algorithm that computes first-order policy gradients through differentiable auction dynamics. The proposed method is evaluated on AuctionNet, where it achieves superior performance and stability compared to standard reinforcement learning baselines.

**Strengths:**

1. Effectively leverages the assumption that the impression sequence is independent of bidding actions to design a more sample-efficient reinforcement learning algorithm.
2. Provides reproducible results with clear implementation details and open-sourced code.
3. Conducts comprehensive experiments on the public AuctionNet benchmark, demonstrating strong empirical performance.

**Weaknesses:**

The assumption that the impression sequence is independent of bidding actions is overly strong and unrealistic. In real-world online advertising systems, auctions are typically conducted in two stages, where advertisers can only observe impressions after entering the second stage (see [1]). Consequently, the observed impression sequence often depends on the advertiser’s bidding behavior. While the paper presents an interesting approach under idealized conditions, its practical contribution remains limited.

[1] Mou et al. "Sustainable online reinforcement learning for auto-bidding." NeurIPS 2022.

**Questions:**

How would the proposed method perform in a more realistic auction setting where the impression sequence depends on the bidding actions?

---

> ### Author Response · Authors · 2025-11-22
> **Rebuttal to Reviewer khCL**
>
> We thank the reviewer for the careful reading and comments, and we address below the concern about our assumption that the impression sequence is independent of bidding actions.
>
> The reviewer questions our assumption that the impression sequence is independent of bidding actions, and referred to [1]. The dependency of impression sequence on bids highlighted by [1] arises because, in some two‑stage advertising systems, only second‑stage impressions are logged while first‑stage selection (which may depend on bids) is hidden, making the **observed** sequence correlated with bids even when the underlying traffic is not. Our work builds on the assumption that the **underlying impression arrival process** $\mathcal I \sim P$ is independent of the advertiser’s bidding policy.  Importantly, this assumption is about the **generation** of the impression stream (user visits, query traffic), not about which subset of that stream is logged or exposed to a specific advertiser.
>
> We agree that in specific architectures where the first‑stage selection depends on bids but is hidden, it is challenging to recover the full impression sequence required by our work. However, in many widely‑used or emerging architectures, it is easy to aquire the instances $\mathcal I$.
>
> - In some large platforms, the first stage is primarily a coarse relevance or CTR‑based pre‑filter that does *not* depend on the specific advertiser’s bid, while auctions and payments are determined only in the second stage; in this case, even when only the second stage is logged, the impression sequence is independent of bids.
> - Smaller or mid‑scale advertising systems employ a single-stage auction per impression;
> - Recent system designs such as UniROM explicitly aim to remove multi‑stage cascades in favor of more unified, end‑to‑end ranking and auction models.
>
> We want to emphasize **the scope and nature of our contribution**. The main contribution of our paper is methodological: we show that, for the step‑level auto-bidding problem, one can exploit impression‑level feedback and high impression density to get a first‑order policy gradient estimator with significantly better performance than standard zeroth‑order RL baselines. This is an algorithmic advance on top of the standard step‑level MDP that has been widely adopted in auto-bidding research. This separation between algorithm design and system integration is analogous to other RL work: state-of-the-art deep RL methods are typically developed and trained in idealized (simulated) environments, and the details of sim‑to‑real transfer or system logging biases are handled in separate engineering layers and follow‑up research.  Our paper is positioned at the algorithmic layer. Thus, the evaluation of our work should not be merely based on direct applicability to some specific ad platform architectures.
>
> The reviewer asks **how FOB would perform in a setting where the impression sequence itself depends on the bidding actions**.  At a conceptual level, if the distribution $P$ depends on the policy, then the true gradient $\nabla J = \nabla \mathbb{E}_{I\sim P_\theta} [J_\theta(I)]$  has an additional term capturing how the policy affects the instance distribution, beyond the pathwise derivative of $J_\theta(I)$ for a fixed $I$. This is related to a recent line of work on performative prediction [3, 4]. Our current estimator explicitly targets $\mathbb{E}_{I}[\nabla_\theta J_\theta(I)]$ under a fixed logging distribution, which remains the correct gradient for optimizing performance on that distribution, and is useful in methods like [3]. More generally, one can compose FOB with online data collection frameworks such as SORL [1]: for example, SORL can be used to safely collect exploration data in the full RAS, while FOB serves as the inner-loop policy optimizer. A thorough theoretical and empirical treatment of the fully coupled case $P_\theta(I)$ is beyond the scope of this paper, but we agree it is an important and interesting direction.
>
> We believe that the concerns raised are important for future system-level extension, but are not grounds for rejection of the algorithmic advance presented in this paper.
>
> [1] Mou et al. "Sustainable online reinforcement learning for auto-bidding." NeurIPS 2022.
>
> [2] Qiu et al. UniROM: Unifying Online Advertising Ranking as One Model, CIKM 2025.
>
> [3] Izzo et al. How to Learn when Data Reacts to Your Model: Performative Gradient Descent, ICML 2021.
>
> [4] Performative Prediction: Past and Future. Hardt et al.

---

> ### Comment · Reviewer_khCL · 2025-11-28
>
> Thank you for your response.
>
> After reading your response, while I still have some concerns regarding the idealized assumption that the impression sequence is independent of the bidding actions, I agree that the assumption holds when the first-stage selection is independent of the bidding policy (it would be helpful if you could provide some examples or relevant literature), or in single-stage advertising settings such as UniROM. Based on this understanding, I will raise my score accordingly.

---

### Official Review · Reviewer_W3J8 · 2025-10-28

**Soundness:** 3
**Presentation:** 2
**Contribution:** 2
**Rating:** 4
**Confidence:** 3

**Summary:**

This paper leverages the high-intensity nature of impression arrivals in step-level real-time bidding to observe the cost and reward functions continuous and smooth. Building on this, it applies Myerson’s lemma to derive Proposition 4.1, which establishes the relationship between the first-order derivatives of cost and reward. The smoothness property is then exploited to approximate these derivatives, and the Adam optimizer is employed during backpropagation to update deep neural networks for bidding. This FOB approach enhances the stability of reinforcement learning training. As a result, the proposed method achieves approximately 82% of the optimal cumulative reward obtained from hindsight linear programming (Equation 1).

**Strengths:**

1.	The authors utilizes the classic results from the auction theory to improve reinforcement learning, resulting in a novel method.
2.	The code is well written and very readable.

**Weaknesses:**

Please refer to the question section.

**Questions:**

1.	In line 231, why does the state information $S_t$ does not include the past second highest price $p_t$? Is it $\gamma_{\tau}$ instead of $v_{\tau, i}$?
2.	Sufficiency of the state space: Theorem 3.1: it is not realistic to assume $\gamma_1, \dots \gamma_T$ to be mutually independent. At least, they are temporally correlated.
3.	Regarding Question 2, how would your experiment results differ if you enlarge the state space to include all the observed information in the past?
4.	As the author mentioned in the limitation, it is not realistic to assume the true value $v_{t,i}$ is observable. Consider, if we can only observe a noisy but unbiased version $\hat{v}_{t, i}$ of $v_{t,i}$, how will it affect your experiment results?
5.	Just curious, why the ``evaluate`` method in class Actor(nn.Module) in fob.py, ppo.py, sac.py, td3.py and uscb.py are slightly different? A follow up question, are the experiment results robust to Actor specification, i.e. rank consistent?
6.	More baseline experiments are suggested, such as online linear LP evaluated in AuctionNet: A Novel Benchmark for Decision-Making in Large-Scale Games baseline program. I saw the implementation of OnlineLpBiddingStrategy. But why are its results not included in the experiment sections?

The reviewer is willing to reconsider the rating if all the above questions are properly addressed.

---

> ### Author Response · Authors · 2025-11-22
> **Rebuttal to Reviewer W3J8**
>
> We thank Reviewer W3J8 for the thoughtful and encouraging review, and for highlighting both the novelty of combining auction theory with RL and the quality of our code. Below we address each question and concern in turn.
>
>
> **1. “Why does the state information not include the past second highest price?”**
>
> In practical auto‑bidding systems, different pieces of auction information become available on different time scales due to data processing and logging delays. Concretely:
>
> - Impression values $v_{t,i}$  are observed **immediately** when the request arrives, via the platform’s real‑time prediction modules.
> - For winning impressions, the corresponding costs (second‑highest prices) are observed **right after** each auction clears, because they are needed to update the advertiser’s budget.
> - For losing impressions, the second‑highest prices are typically only available **after the episode** (e.g., at the end of the day), once the platform has finished aggregating and flushing auction logs.
>
> Under this delayed‑logging view, it is natural to treat losing‑price information as part of the *offline* instance used for training, rather than as part of the online state observed at each step. If one ignores data‑return delays, then in principle one could include $\gamma_\tau$ directly in the state. However, this is only theoretically possible; in practice such a state would be prohibitively high‑dimensional, as discussed in our responses to Questions 2 and 3 below.
>
>
> **2. “Sufficiency of the state space / independence of $\gamma_1,\dots,\gamma_T$” & 3. “How would your results differ if you enlarge the state space to include all observed past information?”**
>
> We agree that in real systems the per‑step instances $\gamma_t$ are not mutually independent; there are temporal correlations. In the paper, we use the 3‑dimensional state $s_t = (t, B_t, B)$ as a minimal abstraction that allows us to clearly present the FOB method. At the same time, FOB as an algorithm is **not restricted** to this particular state; it can be applied with any richer state representation supplied by the practitioner.
>
> Regarding the suggestion to include the *entire* history $\gamma_1 \cdots \gamma_{t-1}$ in the state: while conceptually attractive, this is computationally infeasible in realistic settings. Each $\gamma_\tau$ can encode several thousands of impressions (values, prices), so concatenating all past $\gamma_\tau$ would yield a state with tens or hundreds of thousands of dimensions even for moderate horizons. Training deep RL agents on such a raw, extremely high‑dimensional state is very challenging.
>
> Instead, we explored *moderate* state enlargements using low‑dimensional summaries of recent history:
>
> - **Original (small) state**: $[t, B_t, B]$
> - **Medium state**: $[t, B_t, B, a_{t-1}, r_{t-1}, c_{t-1}]$; which includes last step's action, reward and cost.
> - **Large state**: $[t, B_t, B, a_{t-1}, r_{t-1}, c_{t-1}, \frac 13 \sum_{\tau=t-3}^{t-1} r_{\tau}, \frac 13 \sum_{\tau=t-3}^{t-1} c_{\tau}]$; which adds the average reward and cost of last three steps.
>
> The normalized returns are:
>
> | Budget | FOB (small)   | FOB (medium)  | FOB (large)   |
> | ------ | ------------- | ------------- | ------------- |
> | 150    | 0.819 ± 0.003 | 0.828 ± 0.003 | 0.814 ± 0.007 |
> | 200    | 0.819 ± 0.003 | 0.829 ± 0.002 | 0.814 ± 0.007 |
> | 250    | 0.819 ± 0.003 | 0.829 ± 0.002 | 0.816 ± 0.007 |
> | 300    | 0.821 ± 0.003 | 0.830 ± 0.000 | 0.819 ± 0.006 |
> | 350    | 0.822 ± 0.003 | 0.831 ± 0.001 | 0.822 ± 0.005 |
> | 400    | 0.824 ± 0.003 | 0.833 ± 0.003 | 0.825 ± 0.004 |
>
> These results show that: The medium state that incorporates a small amount of recent history can indeed improve performance slightly over the minimalist state, which aligns with the reviewer’s intuition that ignoring temporal correlation is an approximation. The large state does *not* improve results in our current training regime; its variance is higher and performance is slightly worse, likely because the enlarged state space requires more training epochs and additional hyperparameter tuning to be effective.
>
> We see state design as an **important but orthogonal** research direction; the key point is that FOB is **compatible with any chosen state representation** and benefits from richer states when they can be trained reliably.

---

> > ### Author Response · Authors · 2025-11-22
> > **Rebuttal to Reviewer W3J8 (Cont'd)**
> >
> > **4. “If we can only observe a noisy but unbiased value $\hat{v}_{t,i}$, how does this affect the results?”**
> >
> >  To study robustness to such noise on values, we ran experiments where, at test time, we perturb the values fed into the learned policy: $\hat{v}_{t,i} = v_{t,i} \cdot \text{Uniform}(1-\epsilon, 1+\epsilon),$ so that bids are computed as $a_t \hat{v}_{t,i}$ instead of $a_t v_{t,i}$, with $\epsilon \in \{0.05, 0.1, 0.2\}$.
> >
> > The results are:
> >
> > | Budget | FOB (no noise) | FOB ($\epsilon=0.05$) | FOB($\epsilon=0.1$) | FOB($\epsilon=0.2$) |
> > | ------ | -------------- | --------------------- | ------------------- | ------------------- |
> > | 150    | 0.819 ± 0.003  | 0.817 ± 0.002         | 0.815 ± 0.001       | 0.804 ± 0.001       |
> > | 200    | 0.819 ± 0.003  | 0.818 ± 0.001         | 0.816 ± 0.001       | 0.806 ± 0.001       |
> > | 250    | 0.819 ± 0.003  | 0.820 ± 0.001         | 0.818 ± 0.001       | 0.808 ± 0.001       |
> > | 300    | 0.821 ± 0.003  | 0.821 ± 0.002         | 0.819 ± 0.002       | 0.811 ± 0.002       |
> > | 350    | 0.822 ± 0.003  | 0.823 ± 0.002         | 0.821 ± 0.002       | 0.812 ± 0.002       |
> > | 400    | 0.824 ± 0.003  | 0.825 ± 0.003         | 0.823 ± 0.003       | 0.814 ± 0.002       |
> >
> > FOB is quite robust: With $\epsilon = 0.05$, performance is essentially unchanged. With $\epsilon = 0.1$, the drop is within 0.5%. Even with substantial noise $\epsilon = 0.2$, the drop remains within 2%. These results suggest that FOB can tolerate moderate value‑estimation noise, which is encouraging for deployment with realistic prediction models.
> >
> >
> > **5. “Why are the `evaluate` methods in Actor different across FOB, PPO, SAC, TD3, and USCB? Are results robust to Actor specification (rank consistent)?”**
> >
> > The slight differences in the `Actor.evaluate` implementations simply reflect the standard differences between the underlying algorithms; for all baselines we follow Stable Baselines 3‑style implementations:
> >
> > - **FOB, PPO, SAC** share the same stochastic actor structure. The network outputs a mean and standard deviation, samples a Gaussian action, then applies a `tanh` squashing; in FOB, the `tanh` is written inside an `unnormalized_action` helper, but the semantics match PPO/SAC. SAC additionally computes an entropy‑correction term that requires a log‑probability adjusted for the `tanh` transform.
> > - **TD3 and USCB**: use deterministic actors with *state‑independent* exploration noise, following the original TD3 implementations and also matches Stable Baselines 3.
> >
> > Regarding “rank consistency”: we interpret this as asking whether conclusions about which algorithm performs better are robust to actor‑specification choices, such as using deterministic vs. stochastic policies for FOB. To check this, we ran FOB with a deterministic actor (no sampling, only `tanh` of the mean):
> >
> > | Budget | FOB (stochastic) | FOB_DET (deterministic) |
> > | ------ | ---------------- | ----------------------- |
> > | 150    | 0.819 ± 0.003    | 0.817 ± 0.007           |
> > | 200    | 0.819 ± 0.003    | 0.818 ± 0.005           |
> > | 250    | 0.819 ± 0.003    | 0.820 ± 0.004           |
> > | 300    | 0.821 ± 0.003    | 0.822 ± 0.003           |
> > | 350    | 0.822 ± 0.003    | 0.824 ± 0.002           |
> > | 400    | 0.824 ± 0.003    | 0.827 ± 0.002           |
> >
> > The performance of deterministic and stochastic FOB is very close. Importantly, in all cases FOB remains stronger than the RL baselines reported in the paper, so the relative ranking of methods is stable across reasonable actor specifications.
> >
> >
> > **6. “More baseline experiments, e.g., online linear LP from the AuctionNet benchmark; why are its results not included?”**
> >
> > The Online LP baseline is implemented in the original AuctionNet repository. We did not emphasize comparison with this baseline in the main paper because our primary focus is on developing and evaluating **improved RL methods** tailored for auto-bidding, rather than performing comparisons against heuristic baselines. Nonetheless, we appreciate the reviewer's suggestion and have added the **Online LP** baseline experiments, following the AuctionNet implementation. Its normalized returns versus FOB are:
> >
> > | Budget | OnlineLP | FOB (ours)    |
> > | ------ | -------- | ------------- |
> > | 150    | 0.6566   | 0.819 ± 0.003 |
> > | 200    | 0.6626   | 0.819 ± 0.003 |
> > | 250    | 0.6664   | 0.819 ± 0.003 |
> > | 300    | 0.6664   | 0.821 ± 0.003 |
> > | 350    | 0.6633   | 0.822 ± 0.003 |
> > | 400    | 0.6584   | 0.824 ± 0.003 |
> >
> > FOB consistently and substantially outperforms the online LP baseline across all budget levels.

---

### Official Review · Reviewer_9DbF · 2025-10-31

**Soundness:** 3
**Presentation:** 4
**Contribution:** 3
**Rating:** 6
**Confidence:** 2

**Summary:**

This paper introduces FOB (First-Order policy gradient for auto-Bidding), an algorithm for optimizing agents in online ad auctions. The authors show that standard reinforcement learning methods inefficiently handle the environment's information, overlooking two key properties: (1) availability of impression-level feedback and (2 smoothness of reward/cost functions due to high impression density. FOB exploits this structure by directly computing low-variance, first-order gradients by smoothing historical auction data and backpropagating through the sequential auctions. By leveraging Myerson's lemma to analytically link cost and reward gradients , FOB achieves "superior performance with greater stability and faster convergence" than standard RL baselines on the AuctionNet benchmark.

**Strengths:**

- The paper's primary strength is identifying and exploiting the "nearly differentiable" structure of the auto-bidding problem. Moving from a high-variance, black-box, zeroth-order optimization (standard RL) to a low-variance, first-order optimization by leveraging domain-specific structure is a powerful and well-justified leap.
The application of Myerson's lemma to derive the cost gradient directly from the reward gradient in particular is a clean and insightful theoretical contribution.

- The resulting FOB algorithm is simpler than the deep RL baselines it competes against. The fact that this simpler, more direct method achieves superior performance and stability is a very strong and compelling result.

- The paper includes valuable analyses that strengthen its claims. The ablation on gradient smoothing methods (Table 2) shows the choice isn't critical.

**Weaknesses:**

- The main method assumes full knowledge of the instance $\mathcal{I}$, including the prices $p_{t,i}$ of auctions the agent lost. This is a very strong assumption and often not practical? While this is well-addressed in Appendix C, this limitation should be more prominent in the main paper.

- The method trains by optimizing the expected return over a buffer of historical instances. This implicitly assumes that the distribution $P$ of instances is stationary where s real-world ad auctions, this is often false. Standard online RL methods, while higher variance, may be more adaptive to such non-stationarity.

- The true reward/cost functions $r_t(a)$ and $c_t(a)$ are piecewise-constant step functions, and their true gradient is zero almost everywhere. The paper introduces smoothing (piecewise-linear or SG filter) to create a differentiable surrogate $\tilde{r}_t(a)$. This smoothing introduces an approximation error and a biased gradient. While it works well empirically, the paper lacks a theoretical analysis of this gradient bias or a discussion of its potential impact. *Please correct me if I'm wrong on this!*

**Questions:**

- Why can a user win an auction and then spend up to their budget? Shouldn’t their bid always be capped by the budget?

- Explain the depletion step more and why it is critical to the analysis? Esp compared with just capping bids by the budget.

- In Figure 2, do cumulative costs and rewards converge? Does this imply 0 revenue?

---

> ### Author Response · Authors · 2025-11-22
> **Rebuttal to Reviewer 9DbF**
>
> We sincerely thank the reviewer for the thoughtful evaluation and positive feedback on our work. We are very pleased that you recognized our paper’s key strength, that is, **"identifying and exploiting the nearly differentiable structure of the auto-bidding problem"**. We respond to the weaknesses and questions point by point below.
>
> **The main method assumes full knowledge of the instance.**
>
>   Our method is presented under a full‑feedback assumption in the main text for conceptual clarity, but FOB is designed to handle **both full and partial feedback**. In partial feedback, where only winning prices are observed, we use a one‑sided estimator for the gradients; Appendix C reports that the performance drop compared with full feedback is negligible (within about 0.2%), so this is not a limitation of our method.
>
> **On stationarity of the instance distribution $P$**
>
>   First, our formulation does not assume that impressions **within an episode** (e.g., across 48 steps in a day) are independent (unlike, for example, [1]) or identically distributed: intra‑day non‑stationarity and step‑to‑step fluctuations are fully captured in the instance set. The implicit stationarity assumption is only **across episodes** (e.g., across days) in the replay buffer, which is standard in RL formulations. Concretely, the assumption that $\mathcal I\sim P$ is equivalent to the assumption of a fixed transition kernel $p(s'|
>   s,a)$ in the standard MDP formulation. Cross‑day non‑stationarity is a continual‑RL problem that challenges all methods, **including online RL baselines, rather than a FOB‑specific issue**. In practice, platforms overcome this by routinely refresh the buffer with recent logs or retrain periodically, and FOB is fully compatible with such practices.
>
> **On smoothing, approximation error, and “biased” gradients**
>
>   We appreciate the reviewer's question on the approximation error introduced by smoothing. However, analytically quantifying the “bias” relative to a gradient that is almost everywhere zero is not very informative; instead, a more meaningful question is whether optimizing the surrogate objective gives a (near) optimal solution of the original objective. Consider the piecewise-linear approximation, fix any instance $\mathcal I$, it is easy to show that the gap between approximated reward (cost) and real reward (cost) is small for each step, i.e., $\sup_a|\tilde r_t(a)-r_t(a)| \le \max_i v_{t,i}$ and $\sup_a|\tilde c_t(a)-c_t(a)| \le \max_i p_{t,i}$, where $\max_i v_{t,i}$ and $\max_i p_{t,i}$ bounds the value and price of any single impression, which is negligible. Therefore, optimizing $\sum_{t=1}^T \tilde r_t(a_t)$ subject to $\sum_{t=1}^T \tilde c_t(a_t)\le B$ gives a near optimal solution to the original problem.
>
> **Why can a user win an auction and then spend up to their budget? Shouldn’t their bid always be capped by the budget?**
>
>   In our modeling, bids are generated as $b_{t,i} = a_t v_{t,i}$ without direct dependence on the remaining budget; the budget constraint is enforced at the episode level by stopping the process once cumulative spend reaches $B$.  In real systems, one can equivalently enforce the budget either by this kind of early stopping (once spend hits $B$) or by capping bids using the remaining budget; these are implementation choices at deployment time. Our analysis uses the early‑stopping view because it leads to a cleaner analysis (see the answer to the next question), but the learned policy can be deployed under a bid‑capping implementation.
>
> **Explain the depletion step more and why it is critical to the analysis? Esp compared with just capping bids by the budget.**
>
>   The depletion step is an approximation that exploits the high‑density, small‑price regime: when the remaining budget is insufficient to pay for the next impression at price $p_{t,i}$, we assume the budget is fully depleted at that point and the agent ignores all later impressions in that episode. In reality, a later, cheaper impression might still be affordable, but because each price is tiny compared to the total budget, the contribution of such events is negligible. Conceptually, the depletion step is crucial because it localizes the dependence on the remaining budget $B_t$: almost all steps have rewards and costs that depend **only on the action $a_t$**, and only the single depletion step depends directly on $B_u$, which makes gradient analysis and computation simple. If we instead capped every bid by the remaining budget within each step, many steps could have rewards that depend nontrivially on both $a_t$ and $B_t$, significantly complicating the method. Again, this is a modeling choice for training; a policy trained under the depletion approximation can be deployed in a system that enforces per‑impression bid capping.

---

> ### Author Response · Authors · 2025-11-22
> **Rebuttal to Reviewer 9DbF (Cont'd)**
>
> **On Figure 2 and “convergence” of cumulative costs and rewards**
>
>   Our “reward” is the total value (e.g., clicks, conversions) of the impressions won in that step, whereas “cost” is the total spend (i.e. money); To calculate revenue for the advertiser, one would compute $\text{revenue} = \alpha \cdot \text{value} - \text{cost}$, where $\alpha$ indicates the average revenue the advertiser would make through each click (or conversion). The factor $\alpha$ may vary among different advertisers with different kinds of products. Yet the fact that the revenue may be zero or negative is true: If the advertiser bids too high, the cost may exceed the profit, i.e., $\alpha \cdot \text{value}$.
>
> [1] Balseiro et al., Dual mirror descent for online allocation problems. ICML 2020.

---

### Official Review · Reviewer_hqyD · 2025-11-01

**Soundness:** 3
**Presentation:** 3
**Contribution:** 2
**Rating:** 4
**Confidence:** 4

**Summary:**

The paper introduces an RL based method to optimize budget constrained advertising campaign when the auction is second price.
The algorithm relies on the observations that: the system has access to information that allows for counterfactual estimation; "fluid" approximation is possible: one can compute derivative with respect to the bid multiplier because of the scale involved.
The algorithm is tested against RL baselines on auction Net data and shows superior performances.

**Strengths:**

The algorithms displays superior performance against RL baselines.
The idea of dividing the timeline into different steps allows for counterfactual estimation.
The methodology is simple and could be generalized to other settings.

**Weaknesses:**

- the fact that the highest bid and the value are observed should be explained. Is it because the platform sees everything and the value is derived from a machine learning prediction? Related to that, the experimental setup should be better explained: what is the data made off (columns of the datasets, etc...).
- is RL the right tool? I am surprised the experiment does not include other baselines such as PID, basic heuristics and no regret approaches
- the paper does not provide any theoretical contribution, and it feels like it is overcomplicating some aspects (for example, invoking Myerson's lemma for the second price auction).
- link between J and the derivative of its approximation would be appreciated
- Looking at the experimental setup it is not clear if the problem is  MARL or RL because the authors mention multiple players.
- The realism of the assumption should be better discussed, for instance,

**Questions:**

see weakness

---

> ### Author Response · Authors · 2025-11-22
> **Rebuttal to Reviewer hqyD**
>
> We thank Reviewer hqyD for the valuable comments. We appreciate your positive feedback on the superior performance of our algorithm, as well as the simplicity and potential generality of the approach. Below we respond to each concern in turn.
>
> **"The fact that the highest bid and the value are observed should be explained."**
>
> We emphasize that our First-Order policy gradient for auto-Bidding (FOB) can operate **under both full‑feedback and partial‑feedback settings**. In the main paper, we present the method under full feedback for clarity of exposition; Appendix C then shows that, under partial feedback, the performance degradation is negligible (within about 0.2% in normalized return).
>
> Concretely, when an impression arrives, the agent observes its value $v_{t,i}$ before bidding, provided by real‑time prediction modules (e.g., CTR/CVR models) deployed on the platform. At the end of step $t$, the agent observes the prices $p_{t,i}$ of all *winning* impressions because these are needed for  budget updates. Under full feedback, we additionally assume that the platform reveals the prices of *losing* impressions at the end of an episode, which allows the learner to reconstruct the full instance $\mathcal I$. This setting is realistic when the platform itself trains the bidding agent (as is common in practice), since the platform has full visibility into all auction outcomes, including losing bids. Under partial feedback, where only prices of winning impressions are available, FOB replaces the two‑sided gradient estimator with a one‑sided estimator based on the observed breakpoints to the left of the current action. Appendix C describes this construction in detail and reports experiments showing that partial‑feedback FOB closely tracks full‑feedback FOB across all budget levels.
>
> **"What is the data made of (columns of the datasets, etc.)?"**
>
> The dataset consists of a collection of  **instances** $\mathcal I = (\gamma_1, \ldots, \gamma_T)$. Each $\gamma_t = (n_t, \{v_{t,i}\}_{i=1}^{n_t}, \{p_{t,i}\}_{i=1}^{n_t})$ represents all impressions arriving in time step $t$. Concretely, each row in the dataset corresponds to a single impression and contains four fields: [instance index, step index, value, price]. An excerpt of the raw data looks as follows:
>
> | Instance | Step | Value                  | Price               |
> | -------- | ---- | ---------------------- | ------------------- |
> | 0        | 0    | 0.0007838815431733843  | 0.08436343960707278 |
> | 0        | 0    | 0.0007423389450774249  | 0.07322299402993719 |
> | ...      | ...  | ...                    | ...                 |
> | 0        | 47   | 0.00036851882446571664 | 0.03994214299181739 |
>
> **"Is RL the right tool? I am surprised the experiment does not include other baselines such as PID, basic heuristics, and no-regret approaches."**
>
> The auto-bidding problem is a sequential decision-making problem in a stochastic environment, which makes it a natural fit for RL formulations. Prior work has already demonstrated RL's advantages in this domain (surveyed in our Appendix B).
>
> We appreciate the suggestion to include comparisons with PID controllers, heuristics, and no-regret approaches, and we have conducted such comparisons in our setting:
>
> * PID: A PID controller that follows a target spending plan solved on the training set.
> * OnlineLP: On the training set, solve an optimal bidding parameter for each (remaining time, remaining budget) pair. At test time, directly look up and use these parameters. Aligned with the AuctionNet implementation.
> * DMD (Dual Mirror Descent [6]): A no-regret approach for budget constrained bidding.
>
> | Budget | PID    | OnlineLP | DMD    | FOB (ours)  |
> | :----- | :----- | :------- | :----- | ----------- |
> | 150    | 0.7376 | 0.6566   | 0.7844 | 0.819±0.003 |
> | 200    | 0.7520 | 0.6626   | 0.7965 | 0.819±0.003 |
> | 250    | 0.7641 | 0.6664   | 0.8037 | 0.819±0.003 |
> | 300    | 0.7722 | 0.6664   | 0.8087 | 0.821±0.003 |
> | 350    | 0.7800 | 0.6633   | 0.8121 | 0.822±0.003 |
> | 400    | 0.7869 | 0.6584   | 0.8147 | 0.824±0.003 |
>
> Our FOB algorithm, consistently outperform both PID control and no-regret methods in normalized campaign return. Moreover, these results were obtained with a simple, low-dimensional state representation for FOB and a moderate dataset size; in production, RL methods can take further advantage of high-dimensional features such as campaign history and context, which classical approaches cannot readily exploit.
>
> We appreciate the reviewer's question but emphasize that the main goal of our paper is to develop an improved RL method tailored to the auto-bidding problem, rather than to argue for RL's superiority over all alternative approaches.

---

> ### Author Response · Authors · 2025-11-22
> **Rebuttal to Reviewer hqyD (Cont'd)**
>
> **"The paper does not provide any theoretical contribution"**
>
> Our focus in this work is primarily **practical**—the effectiveness of FOB is validated mainly through experiments. At the same time, **the design of FOB is theoretically supported**: The correctness of our policy‑gradient estimator is rooted in classical policy‑gradient theory (pathwise/first‑order estimators). The analytical relationship between reward and cost gradients, $\nabla_a \tilde{c}_t = a \nabla_a \tilde{r}_t$, follows from Myerson’s lemma (Proposition 4.1), which ensures that our cost gradients are consistent with truthful single‑parameter auctions rather than being ad‑hoc. Our minimalist state design $s_t = (t, B_t, B)$ is theoretically justified via model‑irrelevance abstraction (Theorem 3.1), showing that this reduced state preserves optimality under the stated assumptions.
>
>  **"it feels like it is overcomplicating some aspects (for example, invoking Myerson's lemma for the second-price auction)."**
>
> We respectfully disagree with the comment that invoking Myerson's lemma is overcomplicating. In fact, Myerson's lemma provides critical theoretical grounding for our method. By Myerson's lemma, we establish $\nabla_a \tilde{c}_t = a \nabla_a \tilde{r}_t$ (Proposition 4.1). Although for single-slot second price auctions, this result can be dirived directly using the explicit allocation and payment rules, Myerson's lemma tells us that this relationship is **not a coincidence**. It is a fundamental property that holds for **any single-parameter truthful auction**, including multi-slot truthful auctions [1], boosted second price auctions [2], affine maximizer auctions [3], etc.
>
> **"Link between $J$ and the derivative of its approximation would be appreciated."**
>
> Section 4.2 of the main paper already provides the derivation of the first‑order estimator. We are happy to provide more details if any question remains.
>
> **"Looking at the experimental setup, it is not clear if the problem is MARL or RL because the authors mention multiple players."**
>
> This is a single-agent RL problem. While multiple bidders participate in each auction, we focus on optimizing the policy of one focal advertiser (the agent), treating all other bidders as part of the stochastic environment. This is the standard viewpoint in the auto‑bidding literature [4–8]: the competing bids are modeled as exogenous random variables generated by a stationary distribution that reflects the aggregate behavior of many small agents.
>
> This assumption can be justified by mean‑field models [9], in which each individual advertiser faces a stationary “bidding landscape” generated by a large population of competitors. In large online advertising markets, no single advertiser has enough market power to substantially affect the overall bid distribution, making the single‑agent approximation reasonable and widely used
>
> **"The realism of the assumption should be better discussed."**
>
> This comment appears incomplete, but we interpret it as requesting more thorough discussion of our modeling assumptions. The realism of our assumptions, i.e., randomness only from environment, and high impression density, is discussed in Section 4.1. We are happy to explain more details if any question remains.
>
> [1] Aggarwal et al., Autobidding with Constraints. WINE 2019
>
> [2] Golrezaei et al., Boosted Second Price Auctions: Revenue Optimization for Heterogeneous Bidders, KDD 2021
>
> [3] Curry et al., Differentiable Economics for Randomized Affine Maximizer Auctions. IJCAI 2023.
>
> [4] He et al., A unified solution to constrained bidding in online display advertising. KDD 2021.
>
> [5] Wu et al., Budget constrained bidding by model-free reinforcement learning in display advertising. CIKM 2018.
>
> [6] Balseiro et al., Dual mirror descent for online allocation problems. ICML 2020.
>
> [7] Balseiro et al., Learning in repeated auctions with budgets: Regret minimization and equilibrium. Management Science 2019.
>
> [8] Balseiro et al., Robust budget pacing with a single sample.  ICML 2021.
>
> [9] Iyer et al., Mean field equilibria of dynamic auctions with learning. Management Science 2014.

---

> > ### Comment · Reviewer_hqyD · 2025-11-26
> >
> > Thank you for your answers, it improved my appreciation of the paper, I will raise the score.

---

### Author Response · Authors · 2025-12-04
**Author's Summary of Rebuttal Phase Discussions**

Across all reviews, several **strengths** were consistently recognized.

* Reviewers 9DbF and UFLL highlighted that the paper clearly identifies and exploits the structural properties of auto-bidding, moving from high-variance, black-box RL to low-variance, first-order optimization.
* Reviewers hqyD, 9DbF, khCL, and UFLL agreed that FOB achieves superior performance and stability compared with strong RL baselines on AuctionNet.
* hqyD, 9DbF, and UFLL also emphasized that the method is simple, interpretable, and potentially generalizable to other auction settings.
* W3J8 and khCL explicitly praised the code quality and clarity of the implementation.

During the rebuttal phase, we **addressed the main concerns** raised by the reviewers with additional clarification and experiments. Regarding the “full-information feedback” assumption (raised by hqyD, 9DbF, UFLL), we clarified that FOB supports both full-feedback and partial-feedback regimes and does not rely critically on access to losing prices (Appendix C, and new experiments in our rebuttal to UFLL, point 1). On state design and temporal dependence (W3J8), we explained that the minimal 3D state  $(t, B_t, B)$  is chosen for clarity and theoretical analysis, but FOB is compatible with richer state representations (experiments in our rebuttal to UFLL, point 1&2). On generality of FOB (UFLL, khCL), we sketched concrete adaptation paths for non-truthful auctions in our rebuttal to UFLL, and clarified to khCL that while our current analysis assumes an exogenous impression process, FOB can be composed with online data-collection and performative-prediction frameworks in settings where the impression sequence depends on the bidding policy.

After the rebuttal, **three initially critical reviewers expressed a more positive view of the paper**. Reviewer hqyD stated that our answers “improved my appreciation of the paper” and raised the score; UFLL described the rebuttal as “exhaustive,” and increased both score and confidence while calling it “a good work”; khCL, although still noting the independence assumption as idealized, acknowledged that our clarifications make the contribution more significant and indicated willingness to raise the score.

---

### Meta-Review · Area_Chair_xMko · 2025-12-30

**Summary:**

1.The fact that the highest bid and the value are observed should be explained, proposed by Reviewer hqyD.

2.Strong assumption that full knowledge of the prices of auctions the agent lost, proposed by Reviewer 9DbF and Reviewer UFLL.

3.Lack of baselines, such as OnlineLp, proposed by Reviewer W3J8.

4.The assumption that the impression sequence is independent of bidding actions is overly strong and unrealistic, proposed by Reviewer khCL.

**Reviewer Concerns:**

1.The fact that the highest bid and the value are observed should be explained, proposed by Reviewer hqyD: during the rebuttal phase, the  authors claim that the setting is realistic when the platform itself trains the bidding agent, since the platform has full visibility into all auction outcomes, including losing bids. However, it is not practical as all advertisers can not know the bid of other advertisers during the bidding phase. Thus this point is still outstanding.

2.Strong assumption that full knowledge of the prices of auctions the agent lost, proposed by Reviewer 9DbF and Reviewer UFLL: during the rebuttal phase, the authors claim that the performance drop is negligible without these informations. However, it is not practical as all advertisers can not know the bid of other advertisers during the bidding phase. Thus this point is still outstanding.

3.Lack of baselines, such as OnlineLp, proposed by Reviewer W3J8: during the rebuttal phase, the authors compare FOB with onlineLP. However, the paper does not discuss and compare stoa bidding baselines, such as diffusion-based and decision transformer-based bidding baselines. Thus this point is still outstanding.

4.The assumption that the impression sequence is independent of bidding actions is overly strong and unrealistic, proposed by Reviewer khCL: during the rebuttal phase, the authors claim that the paper builds on the assumption that the underlying impression arrival process
 is independent of the advertiser’s bidding policy. However, it is not practical. Thus this point is still outstanding.

**Reviewer Scores:**

Reviewer hqyD would increase his or her score from 4 to 6 if he or she has been able to participate fully in the discussion.

Reviewer 9DbF would keep his or her score as 6 if he or she has been able to participate fully in the discussion.

Reviewer W3J8 would keep his or her score as 4 if he or she has been able to participate fully in the discussion.

Reviewer khCL would keep his or her score as 2 if he or she has been able to participate fully in the discussion.

Reviewer UFLL would increase his or her score from 4 to 6 if he or she has been able to participate fully in the discussion.

---

> ### Public Comment · ~Haoming_Li2 · 2026-02-28
> **Authors' Response to Meta Review**
>
> We thank the Area Chair for the summary. However, we would like to respectfully point out a few misunderstandings and factual errors in the Meta Review regarding our paper's setting and the reviewers' comments. As a result, some of the listed reviewer concerns are based on an incorrect understanding of our work and are not applicable.
>
> 1. Regarding Reviewer Concerns 1 & 2
>
> The meta review states: "it is not practical as all advertisers can not know the bid of other advertisers during the bidding phase."
>
> This is a misunderstanding of our setting. We have **never** assumed that a bidder can observe other bidders' bids during the bidding phase. Throughout the paper, we assume that the bidder can only observe the value during bidding (Section 3, line 189). Our full feedback assumption (Page 5, footnote 2) refers to feedback in hindsight: the agent observes the highest competing bid **after** an auction concludes. Furthermore, as detailed in Appendix C, we have demonstrated that FOB still works beyond the full feedback assumption.
>
> 2. Regarding Reviewer Concern 3
>
> The meta review mentions: "the paper does not discuss and compare stoa bidding baselines, such as diffusion-based and decision transformer-based bidding baselines."
>
> The diffusion-based and decision transformer-based baselines mentioned are both **offline** RL methods, whereas our paper proposes an **online** RL method. These represent two entirely different training paradigms that rely on **different training data**. Comparing them directly is not reasonable. For example, offline RL methods heavily depend on the quality of actions in the training data, while our method is completely independent of historical actions.
>
> 3. Factual Error Regarding Reviewer Scores
>
> The Area Chair's summary contains a factual error regarding the reviewers' scores. The meta review states: "Reviewer khCL would keep his or her score as 2 if he or she has been able to participate fully in the discussion."
>
> However, reviewer khCL explicitly stated in an official comment: "Based on this understanding, **I will raise my score accordingly.**"
>
> We provide these clarifications here to correct the factual record for future readers and to ensure our problem setting and the reviewers' original feedback are accurately represented.

---

### Decision · Program_Chairs · 2026-01-26

Reject